# Design and analysis of a wake model for spatially heterogeneous flow

Alayna Farrell[1], Jennifer King[1], Caroline Draxl[1], Rafael Mudafort[1], Nicholas Hamilton[1], Christopher J. Bay[1], Paul Fleming[1], and Eric Simley[1]

[1]National Renewable Energy Laboratory, Golden, CO, 80401, USA

**Correspondence:** afarrell@msu.edu, floris@nrel.gov

**Abstract.** Methods of turbine wake modeling are being developed to more accurately account for spatially variant atmospheric conditions within wind farms. Most current wake modeling utilities are designed to apply a uniform flow field to the entire domain of a wind farm. When this method is used, the accuracy of power prediction and wind farm controls can be compromised depending on the flow-field characteristics of a particular area. In an effort to improve strategies of wind farm wake modeling and power prediction, FLOw Redirection and Induction in Steady State (FLORIS) was developed to implement sophisticated methods of atmospheric characterization and power output calculation. In this paper, we describe an adapted FLORIS model that features spatial heterogeneity in flow-field characterization. This model approximates an observed flow field by interpolating from a set of atmospheric measurements that represent local weather conditions. The objective of this method is to capture heterogeneous atmospheric effects caused by site-specific terrain features, without explicitly modeling the geometry of the wind farm terrain. The implemented adaptations were validated by comparing the simulated power predictions generated from FLORIS to the actual recorded wind farm output from the Supervisory Control And Data Acquisition (SCADA) recordings and large eddy simulations (LES). When comparing the performance of the proposed heterogeneous model to homogeneous FLORIS simulations, the results show a 14.6% decrease for Mean Absolute Error (MAE) in wind farm power output predictions for cases using wind farm SCADA data, and a 18.9% decrease in LES case studies. The results of these studies also indicate that the efficacy of the proposed modeling techniques may vary with differing site-specific operational conditions. This work quantifies the accuracy of wind plant power predictions under heterogeneous flow conditions and establishes best practices for atmospheric surveying for wake modeling.

## 1 Introduction

Low-fidelity wake modeling utilities such as FLOw Redirection and Induction in Steady State (FLORIS) are typically used for the estimation of wind farm power output or the implementation of wind farm controls that help improve the overall performance of a wind farm. This includes implementing real-time corrective strategies that aid in reducing stress-inducing loads on turbines (Boersma et al., 2017), avoiding operational side effects like noise pollution (Leloudas et al., 2007) or shadow flicker (Clarke, 1991), and maximizing power output through methods of wake steering and power grid optimization (Fleming et al., 2017b). FLORIS, and most other controls-oriented wake modeling utilities, implement advanced wake modeling algorithms that are capable of producing accurate results in a uniform set of atmospheric conditions (Fleming et al., 2019). However, the accuracy of any wake model is highly dependent on its ability to recreate the characteristics present. It is important for these

models to be able to emulate the naturally occurring state of the wind farm as closely as possible for the controls processes and power-prediction functionalities to operate with reliable accuracy. Most current controls-oriented wake modeling utilities use a homogeneous approximation to characterize the initial state of the atmosphere, which can introduce major inaccuracies in the simulation of wind farm flow interactions.

The consequences are particularly evident when observing the accuracy of power predictions for wind farms located within complex terrain, or wind farms that are otherwise subject to spatially variant conditions in the atmosphere. Because these atmospheres are subject to dramatic changes in the velocity and direction of wind, it is difficult to anticipate how the resulting wakes will form and what kind of power output should be expected. In Yang et al. (2019), an analysis of the impact of spatial heterogeneity in wind farm flow is presented for a site within complex terrain. This study showed that using averaged values of wind conditions caused short-term wind power forecasting to be less accurate, due to spatial heterogeneity within the wind field and the variability of wind turbine power curves. With these effects considered, the current version of FLORIS and many other wake model utilities are not constructed to accurately model fluid flow under these conditions.

It should be noted that there are existing wake models that incorporate elements of heterogeneous wake effects caused by varying atmospheric conditions. For example, one model presented in You et al. (2016) takes a statistical approach in representing heterogeneous power deficit caused by wind farm-flow interactions in spatially variant weather conditions. Another method discussed in Shao et al. (2019) proposes an interaction model used for calculating the turbulence intensity of overlapping wakes, and represents the relative positions of wind turbines under arbitrary and varying wind direction conditions. Brogna et al. (2020) presents a technique that superimposes the centerlines of wind turbine wakes in complex terrain by following the streamlines of the background flow field. Clustering methods have also been implemented, such as Katic et al. (1986) and Clifton and Lundquist (2012), where the turbines of a wind farm are sectioned into groups, assigning differing atmospheric characteristics to each cluster of turbines to mimic the heterogeneous conditions observed in natural atmospheres. Additionally, many approaches implement data-driven wake model correction parameters to achieve more accurate solutions, such as those proposed by Schreiber et al. (2019); Shapiro et al. (2019); Howland et al. (2020); Teng and Markfort (2020).

The aforementioned models present many methods for approximating farm-flow interaction in heterogeneous conditions. As a contribution to this area of research, this article will present a modified version of FLORIS that features an advantageous capability in modeling wind farms with spatially variant weather conditions and complex terrain. This adapted version of FLORIS presents several novel developments within the scope of controls-oriented wake modeling research: an interpolation algorithm is implemented, which allows the user to define a gradient of atmospheric characteristics across the flow field, based on several measurements within or adjacent to the wind farm; elements of spatially variant wind direction, wind speed, and turbulence intensity are integrated into wake calculations of the preexisting FLORIS model; and an additional method is introduced to minimize error in power-prediction accuracy caused from high-turbulence intensity and wind speed variance.

The objective in developing this proposed model is to capture a more accurate representation of the effects of wind farm wake interactions within complex terrain without actually resolving any terrain geometry during simulation. This study aims analyze the accuracy of power output predictions and wake modeling performance for the proposed wake model, through comparisons to Large Eddy Simulations (LES) wind farm Supervisory Control And Data Acquisition (SCADA) records.

## 2 Existing FLORIS model

FLORIS (NREL, 2020) is a wake modeling utility that is equipped with tools designed for the control and optimization of wind farms, and is being developed at the National Renewable Energy Laboratory (NREL) in collaboration with Delft University of Technology. This tool uses several computational modeling techniques paired with controls algorithms to approximate and optimize wind turbine wake interactions through integration of real-time Supervisory Control And Data Acquisition (SCADA) data recorded from wind farms. FLORIS implements the concept of steady-state averaging to simulate the observed dynamic behavior within a wind farm for each iteration in time, and can also be used as a simulation tool to compute farm-flow interactions in wind farms under user-defined atmospheric conditions. This section will give an overview of the mathematical theory in which the formulations of the wake models of FLORIS were based. These concepts are also explained in greater detail in Annoni et al. (2018) and Hamilton et al. (2020).

### 2.1 Turbine power-output model

The operation and performance of a turbine is modeled with respect to the relationship between the thrust coefficient, $C_T$, and power coefficient, $C_P$. The dependence between these two terms characterizes a turbine's power output and wake propagation, therefore making the understanding of this relationship fundamental to the design and operation of wind farm controls. To model the performance behaviors of a given turbine, a table is constructed inside of FLORIS that tabulates $C_T$ and $C_P$ with respect to wind speed. This table can be set to a user's self-obtained data, generated independently by NREL's FAST (Jonkman, 2010), or by integrating CCBlade (Ning, 2013) with FLORIS. The relationship between $C_T$ and $C_P$ can also be defined through the concept of actuator disk theory. This theory relates the turbine power output and thrust through the axial induction factor, $a$, which can be calculated using the definitions from Burton et al. (2002) and Bastankhah and Porté-Agel (2016):

$$C_P = 4a(1-a)^2 \tag{1}$$
$$C_T = 4a(1-a) \tag{2}$$

From these values, the power can then be calculated for turbines under steady-state and yaw-misalignment conditions, using the following relationship provided by Burton et al. (2002):

$$P = \frac{1}{2}\rho A C_P u^3 \cos^p(\gamma) \tag{3}$$

where $\rho$ is the air density, $A$ is the rotor-swept area, $u$ is the rotor-averaged wind speed, and $p$ is a tuneable parameter that accounts for the power losses due to yaw misalignment seen in simulations (Burton et al., 2002; Fleming et al., 2017a). Thus far, the turbine model discussed in this section does not consider the effects that turbulence may have on the relationship between power output and wind speed. However, Sheinman and Rosen (1992) analyze the effects of turbulence intensity on wind farm power output. In this study, it is shown that turbine power output can be overestimated by more than 10% if turbulence intensity is not considered. Many empirical and machine-learning methods have been proposed to solve this issue. However, a

nonparametric statistical averaging model may be preferred, such as the model developed in Hedevang (2014). In Section 3.5, a new method of implementing a turbulence-dependent correction to power will be discussed for FLORIS applications.

## 2.2 Velocity deficit

FLORIS provides an option to select particular models for wake velocity deficit and wake deflection separately to suit the user's performance needs. The variety in modeling capabilities reflects a range of trade-offs between computational efficiency and the number of detailed physics applications applied to calculations. If a model is more computationally expensive, it is likely to implement more sophisticated algorithms as well, in hopes of achieving a more accurate result. These models all have a different approach to modeling turbine wake interactions, and offer different strengths and weaknesses in functionality. Most models can either be classified as a velocity deficit, or a wake deflection calculation, but there are also the Gaussian and Curl models that incorporate both calculations and extend further into the overall FLORIS wake modeling structure and control tools. For the purposes of this article, only the Gaussian wake model will be explained in-depth. See Annoni et al. (2018), Martínez-Tossas et al. (2019), and Bay et al. (2019) for details on additional models in FLORIS.

## 2.3 Gaussian wake

The Gaussian Wake Model is comprised from a series of papers, including Bastankhah and Porté-Agel (2014); Abkar and Porté-Agel (2015); Niayifar and Porté-Agel (2015); Bastankhah and Porté-Agel (2016); Dilip and Porté-Agel (2017). This model is a method of calculation that is integrated into the structure of all FLORIS wake modeling and control tools. It integrates the concepts of the Bastankhah and Porté-Agel wake deflection model, the self-similar velocity deficit model, and elements of atmospheric stability into one comprehensive method based off of the concept of a Gaussian wake (Pope, 2000). This section will describe the different concepts that are implemented in this model.

### 2.3.1 Self-Similar Velocity Deficit

The Gaussian model computes the streamwise velocity deficit at any point in a turbine's wake by using analytical formulations of Reynolds-averaged Navier-Stokes (RANS) equations to an assumed Gaussian wake profile. The Gaussian wake is based on the self-similarity theory used for free shear flows (Pope, 2000), and is developed under the assumption of no pressure gradients within the initial undisturbed free-stream flow and uniform flat terrain (Bastankhah and Porté-Agel, 2014). To calculate the velocity deficit, $u(x,y,z)$, behind the rotor of a turbine:

$$u(x,y,z) = U_\infty \left( 1 - C \left[ \exp\left( -(y-\delta)^2/2\sigma_y^2 \right) \cdot \exp\left( -(z-z_h)^2/2\sigma_z^2 \right) \right] \right) \tag{4}$$

$$C = 1 - \sqrt{1 - \frac{(\sigma_{y0}\sigma_{z0})C_0(2-C_0)}{\sigma_y\sigma_z}}$$

$$C_0 = 1 - \sqrt{1 - C_T},$$

where $U_\infty$ is the freestream velocity; $x, y,$ and $z$ represent the spatial coordinates in the streamwise, spanwise, and vertical directions, respectively; and $z_h$ is the turbine hub height. $C$ is the velocity deficit at the wake center; $\delta$ represents the wake deflection computed with equations from Bastankhah and Porté-Agel (2016); and $\sigma$ denotes the wake width in the lateral ($y$), and vertical ($z$) directions. The subscript "0" references a term's initial value at the start of the far wake.

The wake width in the $y$ and $z$ directions, $\sigma_y$ and $\sigma_z$, are determined by the thrust coefficient, $C_T$, and the wake expansion rate, which is parameterized by $k_y$ and $k_z$:

$$\frac{\sigma_z}{D} = k_z \frac{(x - x_0)}{D} + \frac{\sigma_{z0}}{D}, \quad \text{where} \quad \frac{\sigma_{z0}}{D} = \frac{1}{2}\sqrt{\frac{u_R}{U_\infty + u_0}}, \tag{5}$$

$$\frac{\sigma_y}{D} = k_y \frac{(x - x_0)}{D} + \frac{\sigma_{y0}}{D}, \quad \text{where} \quad \frac{\sigma_{y0}}{D} = \frac{\sigma_{z0}}{D}\cos\gamma, \tag{6}$$

where $D$ is the rotor diameter, $u_R$ is the velocity at the rotor, $\gamma$ denotes the turbine's yaw offset, and $u_0$ represents the maximum velocity deficit in the wake. Parameters $k_y$ and $k_z$ are dependent on the value the ambient turbulence intensity, $I_0$, as noted in Eq. 8.

The findings of Abkar and Porté-Agel (2015) demonstrate that $k_y$ and $k_z$ grow at different rates, but in order to simplify the model, $k_y$ and $k_z$ are usually set as equal. The total velocity deficit at any point in the domain of fluid flow can then be calculated by combining the wakes using the sum-of-squares method described in Katic et al. (1986).

In the scope of this study, it is important to note that the introduction of spatial heterogeneity in initial wind conditions (which is a key principle in the proposed model) violates the original assumption of no pressure gradient for the derivation of the Gaussian wake model. Although this limits the model's ability to conserve key principles that govern the physical dynamics of fluid flow, the results of this study show that the measured improvements in model accuracy outweigh the consequences of incomplete conservation. In Brogna et al. (2020), a modified Gaussian wake model is implemented to simulate wind farms in complex terrain, but the spatial $U_\infty$ evolution is considered only in the superposition of wakes and is omitted for the calculation of the velocity itself. The benefits of an approach similar to this could be investigated in future FLORIS developments to improve overall momentum conservation for the heterogeneous model.

### 2.3.2 Atmospheric Stability

The Gaussian model also implements methods proposed by Abkar and Porté-Agel (2015); Niayifar and Porté-Agel (2015), which characterize the effects of atmospheric stability by analyzing the levels of veer, shear, and changes to turbulence intensity in the fluid flow. Stull (2012) discusses that an accurate representation of atmospheric stability requires the measurement of many other variables in the atmosphere; but without detailed recordings of elements such as temperature profiles and vertical flux, the three chosen parameters are able to give a rough idea of the state of the atmosphere in the FLORIS model.

To implement the effects of shear, $\alpha_s$, the power-log law of wind is used to define the initial wind speed in the flow field, $U_{\text{init}}$:

$$\frac{U_{\text{init}}}{U_\infty} = \left(\frac{z}{z_h}\right)^{\alpha_s}, \tag{7}$$

where a high shear coefficient ($\alpha_s > 0.2$) is indicative of stable atmospheric conditions, and a low shear coefficient ($\alpha_s < 0.2$) characterizes an unstable atmosphere (Stull, 2012).

The Gaussian model was designed to avoid the inaccuracies caused by neglecting the effects of turbulence intensity by implementing methods introduced by Niayifar and Porté-Agel (2015). This also includes added turbulence caused by nearby turbine operation to more accurately calculate the rate of wake expansion. Many other linear-flow models use a constant parameter that defines the rate of wake expansion and has no dependency on the operating conditions of the turbine (Jensen (1983)). From the concepts of Niayifar and Porté-Agel (2015), the Gaussian model relates the rate of wake expansion in the lateral and vertical directions directly to the ambient turbulence intensity present at a turbine and two tuned parameters, $k_a = 0.38371$ and $k_b = 0.003678$:

$$k_y = k_z = k_a I + k_b. \tag{8}$$

The turbulence intensity, $I$, is calculated by superimposing the initial ambient turbulence intensity ($I_0$) with the sum of the added turbulence caused by the operation of each influencing upstream turbine, $j$ and $I_j^+$. The following relationship is used in FLORIS to calculate the ambient turbulence intensity at a given turbine with respect to neighboring turbine wakes:

$$I = \sqrt{\sum_{j=0}^{N} \left(I_j^+\right)^2 + I_0^2}. \tag{9}$$

$N$ refers to the number of upstream turbines that create a wake that adds to the ambient turbulence intensity at a downstream turbine's location. In Niayifar and Porté-Agel (2015), this number was assumed to be one, and the closest turbine was only taken into account because it would theoretically give the maximum amount of added turbulence. In the Gaussian model used in FLORIS, all turbines within a distance of $15D$ upstream and $2D$ in the span-wise ($y$) direction are included. Although the saturation effects of turbulence are not yet fully understood in this context, this formulation was shown to be a more accurate method of calculating added turbulence intensity in the findings Chamorro and Porté-Agel (2011), which found that turbulence intensity typically accumulates over two to three turbine rows, but then levels off to an equilibrium at this point.

Based on the original definition proposed in Crespo and Hernández (1996), the following expression in Eqn. 10 has been tuned through comparisons to high fidelity CFD simulations (King et al., 2020b) and several field studies (Fleming et al., 2019, 2020b) to accurately calculate the added turbulence due to upstream turbine $j$:

$$I_j^+ = A_{\text{overlap}} \left(0.5 a_j^{0.8} I_0^{0.1} (x/D_j)^{-0.32}\right), \tag{10}$$

where $D_j$ denotes the diameter of turbine $j$, and $A_{\text{overlap}}$ refers to the fraction of the rotor-swept area of the downstream turbine that intersects with the cross-sectional area of the wake from the upstream turbine. The axial induction factor, $a_j$ is evaluated based on the value of $C_T$, as defined in Burton et al. (2002) and Bastankhah and Porté-Agel (2016).

As noted earlier, the Gaussian wake model was developed under the assumption of flat terrain. Since the heterogeneous model was specifically designed to best benefit wind farms located in complex terrain, it important to know the consequences of violating this assumption. In Fleming et al. (2020b), a field study is presented that focuses on analyzing the performance of the tuned parameters in Eqn. 10, by comparing two campaigns located in comparatively simple and complex terrains. The findings of this study indicate that inaccurate tuning of the tuned variables may worsen FLORIS's typical tendency to underpredict wake losses in areas with complex terrain.

## 3 Changes to the FLORIS model

Previously, FLORIS derived the initial wind speed, wind direction, and turbulence intensity by using one value to represent the entire flow-field domain. In this article, we describe the modifications to FLORIS to accommodate heterogeneous flows. This section will explain the methods used to calculate wakes based on the gradient of values observed in the undisturbed flow field without wake effects. The motivation behind this development was to create a more detailed characterization of the initial state of the atmosphere, which leads to improvements in the power predictions of a wind farm.

### 3.1 Initializing the heterogeneous flow field

To implement heterogeneity in FLORIS, an interpolation is performed based on several input values assigned to spatially varying coordinates inside or adjacent to the wind farm (see Fig. 2a). These initial inputs are used to approximate the value of atmospheric characteristics at the location of every turbine within the wind farm, and at each individual grid point of the FLORIS flow field. FLORIS performs methods of interpolation and extrapolation using software packages provided by SciPy: an open-source scientific computing library for the Python programming language (Virtanen et al., 2020). The packages used in this method include a piecewise linear interpolant and a nearest neighbour interpolant, which are combined to create an algorithm that calculates a unique value for each x and y coordinate within the flow field. Fig. 1 shows a pseudo-code diagram of this process for reference.

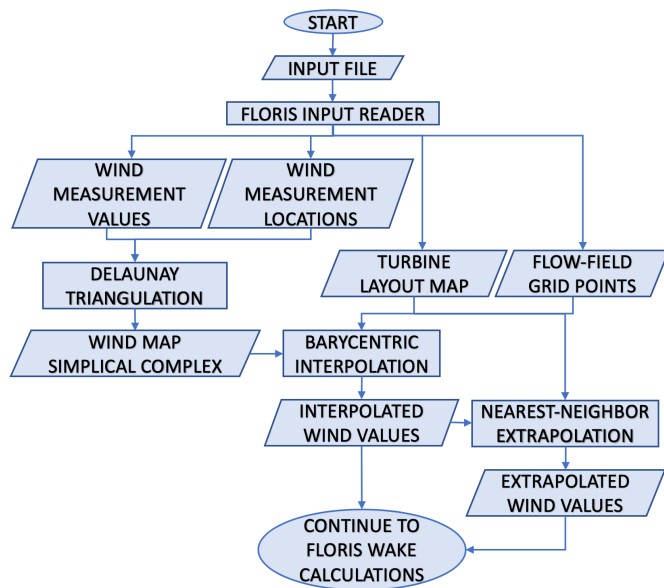

**Figure 1.** A diagram representing the processes performed during the initialization of the heterogeneous FLORIS model.

The process begins with implementing a piecewise linear interpolation method for all points within the region defined by the input coordinates. First, Delaunay triangulation is performed using the Quickhull algorithm discussed in Barber et al. (1996). This method forms triangular connections between input points, based on their relative coordinates, and defines each triangle by ensuring its circumcircle remains empty. The result of this triangulation generates a mesh of triangular elements called a simplicial complex. Further details on the concept of Delaunay triangulation are explained in-depth in Shewchuk (1999) and

Barber et al. (1996).

The next step in determining the interpolated values is to use the established triangular elements to perform barycentric interpolation. During this step, the barycentric coordinates of each point of interest are determined relative to the triangular element in which it resides. Based on each set of barycentric coordinates, the interpolated result is calculated using a weighted average of the values defined at the triangle's vertices (Floater, 2015). A visual depiction of the methods utilized in this

piecewise interpolation method (Delaunay Triangulation and Barycentric Interpolation) are shown in Fig. 2b. After these processes are complete, FLORIS assigns the interpolated values to each flow-field grid point and turbine location inside the triangulated region bounded by the input coordinates. Any points that fall outside of this region must be determined through additional extrapolation processes.

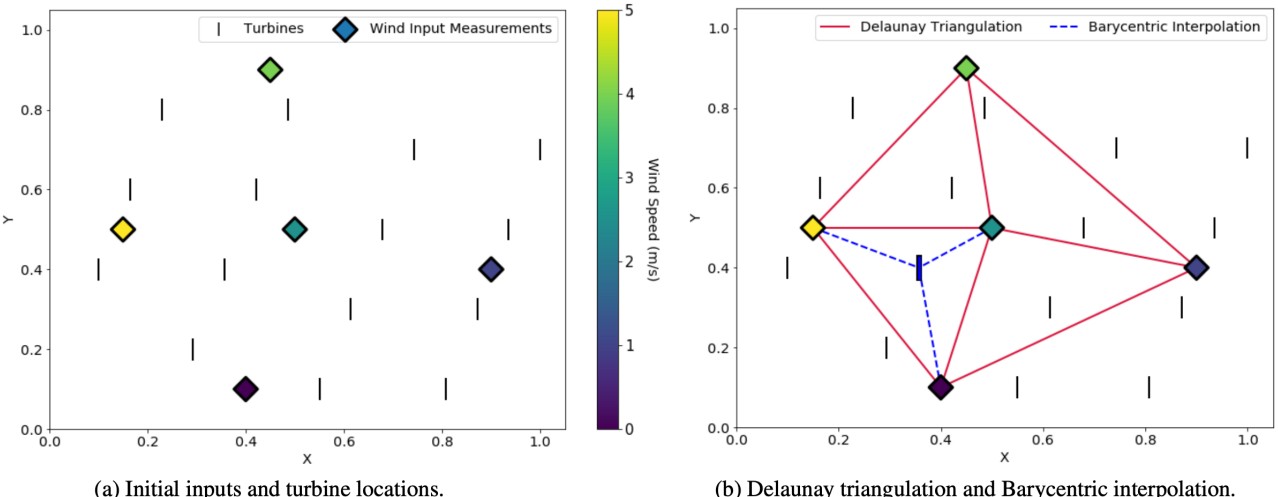

(a) Initial inputs and turbine locations.

(b) Delaunay triangulation and Barycentric interpolation.

**Figure 2.** A visual depiction of the methods used interpolate and define atmospheric characterization values at specific points within the input coordinates.

Linear Barycentric interpolation was chosen to implement for this step because it is relatively efficient in computation and can be easily implemented without requiring any input parameters other than the locations and values of wind measurements. Although it must be noted that the accuracy of the interpolated values is dependent on the quality of input measurements provided, the complexity of the terrain geometry, and the weather patterns observed in the physical wind farm.

The extrapolation process implements a nearest-neighbor interpolant to calculate all remaining unknown values. Using the recently interpolated point values in addition to the original input values, this method operates by selecting a single value at the nearest location to the point being extrapolated, and assigning this nearest value to the extrapolated point. A visualization of this calculation is depicted in Fig. 3.

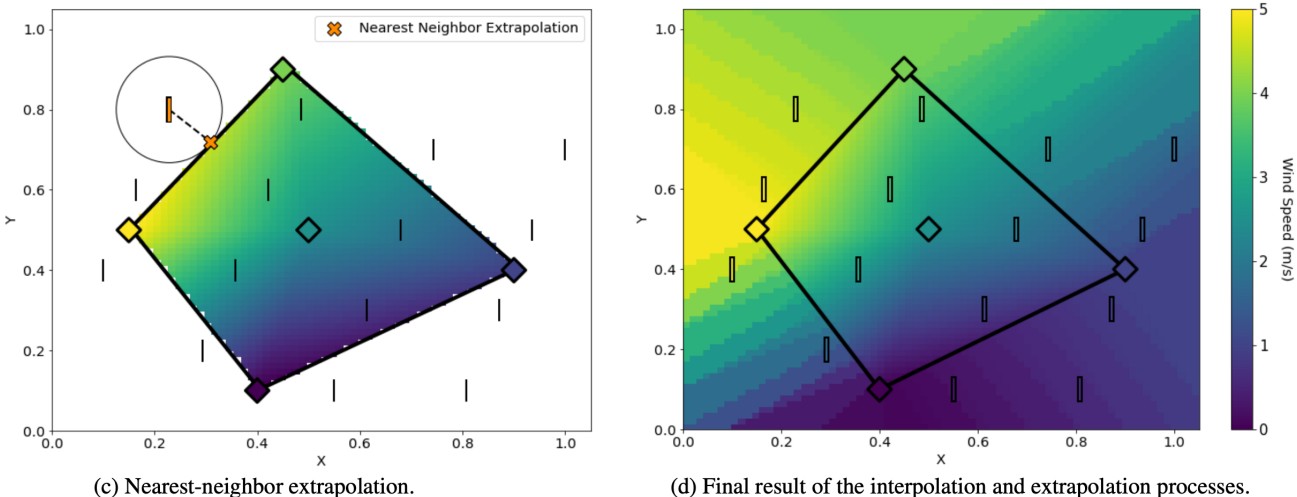

(c) Nearest-neighbor extrapolation.

(d) Final result of the interpolation and extrapolation processes.

**Figure 3.** The extrapolation process used to define the remaining values for characterization of the initial state of fluid flow.

The nearest-neighbor extrapolation method was chosen because it defines a feasible relationship between input measurements and does not attempt to extrapolate using a formula derived from a curve-fitting or trend-predictive algorithm. Many other extrapolation methods attempt to predict a rate of change outward of the interpolation domain by implementing a function that approximates a predicted progression of extrapolated values. For example, it was found that the analytic continuation of Radial Basis Functions (RBF) and fitted polynomial splines outside of the initial domain often produced a non-feasible output that did not respect the physical limitations of the atmospheric characteristic being extrapolated. Although it was speculated that these methods could likely be adjusted with tuning factors to fit extrapolated data within feasible bounds, efforts to do this were not explored in this study. Instead, the nearest-neighbor algorithm was chosen to simplify implementation of realistic extrapolation within the model.

When solving for the interpolated and extrapolated values for turbulence intensity and wind speed, values are easily computed because they are defined by values on a non-cyclical scale. Because wind direction is represented using angles in degrees, the interpolation and extrapolation methods must be circular. The issue of interpolating circular data was addressed by simply computing the interpolation twice for each angle of wind direction, $\Phi$: once for the cosine component, $\alpha$, and again for the sine component, $\beta$. The wind direction in a wind farm, $\Phi$, can be defined as:

$$\Phi = \arctan 2 \left( \frac{\beta}{\alpha} \right) \tag{11}$$

Where $\alpha = \cos \Phi$, and $\beta = \sin \Phi$. After $\Phi$ is computed, the wind direction interpolation can then be defined for the entire wind farm.

It should be noted that the vertical (z) dimension is not considered when interpolating and extrapolating from the atmospheric inputs. Instead, all input values are assumed to be at the same z location, and the interpolation is performed on a two-dimensional plane at this height. Although this approximation may result in a less accurate result, this approach allows the interpolation and extrapolation algorithm to operate with less computational cost. Differences in wind speed due to variations in turbine hub height are calculated using the power law in Eq. 7, as described in subsection 3.2.

### 3.2 Heterogeneous wind speed

Before FLORIS performs any calculations for velocity deficit in wakes, it first assigns an initial value of wind speed ($U$) to each grid point in the flow-field grid. In a homogeneous case, these grid points would all have the same value across an $x - y$ plane, but in a heterogeneous case, these grid points all have different values, dependent on the initial values that have already been established through interpolation. After $U$ is defined at each grid point, the wind-speed values at each $x$, $y$, and $z$ coordinate in the flow-field domain are defined as $U_{init}$, calculated using the power law in Eq. 7. From this point, the calculation of wakes proceeds in the same way as the homogeneous cases, with the exception of a more complex algorithm for accounting for changes in wind direction, as explained in Section 3.3. The velocity deficit behind each turbine is calculated by applying Eq. 4 from Section 2.3.1, where the free-stream velocity ($U_\infty$) in Eq. 4 is defined as the local $U_{init}$ values at each flow-field

grid point. Figure 4 shows visualizations of the resulting wakes after subtracting the calculated velocity deficit from the initial free-stream velocity at each flow-field grid point.

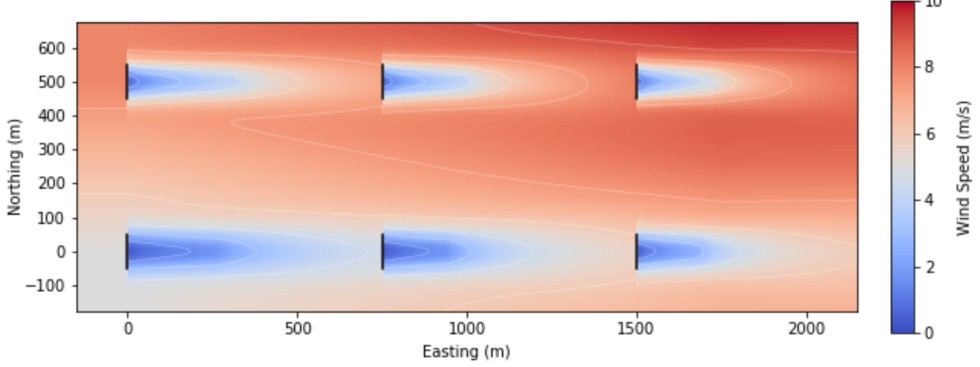

(a) Horizontal plane of the FLORIS simulation, taken at the turbine hub height (90m).

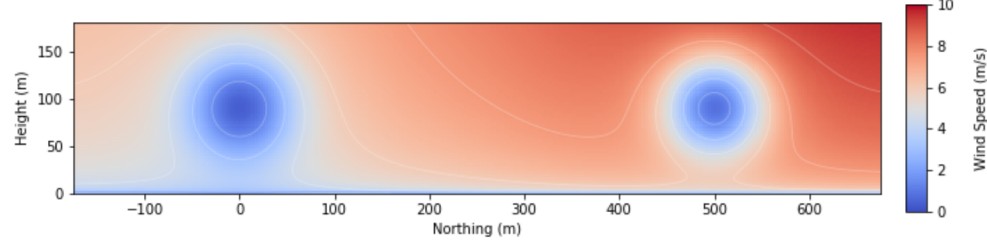

(b) Vertical cross-plane of the FLORIS simulation, taken at 760m east of the origin (10m downstream of middle turbines).

**Figure 4.** Visualizations of two planes showing the FLORIS flow field during a simulation with heterogeneous wind speed.

### 3.3 Heterogeneous wind direction

Similar to wind speed, an interpolation of wind direction is initially established across the flow-field grid through the methods of interpolation discussed in Section 3.1. The input values of wind direction are defined so that 270 degrees represents wind movement from west to east (see Fig. 5a), then once FLORIS begins computations with these wind directions, the values are converted so that 0 degrees represents the wind traveling from west to east (see Fig. 5b). Using these wind direction values, the turbine coordinates are rotated about the center of the simulation domain at these angles, as exemplified in Fig. 5b.

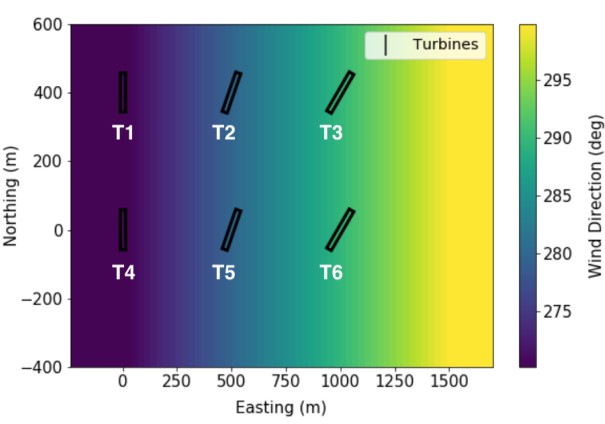
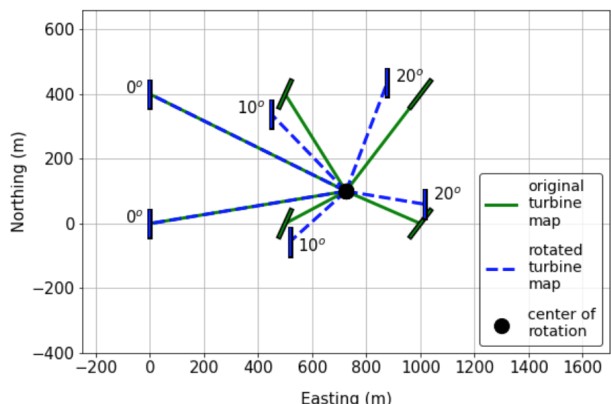

(a) Interpolated values of wind direction, defined at every point within the flow-field domain.

(b) Rotation of turbine locations relative to the center of rotation.

**Figure 5.** A depiction of the initial processes before the calculation of wakes. Figure 5a shows the result of wind direction interpolation, and Fig. 5b shows the process used to define the location of the rotated turbine map. The turbines will be referred to individually as T1, T2, ... T6, as defined in Fig. 5a.

Using the rotated turbine map shown in Fig. 5b for reference, the flow field is adjusted to calculate each turbine wake independently, starting with the turbine that is the furthest upstream. To initiate the rotation of the flow-field grid, the grid points are rotated to the angle that is defined at the given turbine. This initial step is exemplified in Fig. 6 for the calculation of the velocity deficit behind turbine T6 only, but this will also be repeated for each turbine in the entire wind farm. This step

is necessary to put the original non-rotated grid points in a frame of reference relative to the each specific turbine as their particular wake is calculated.

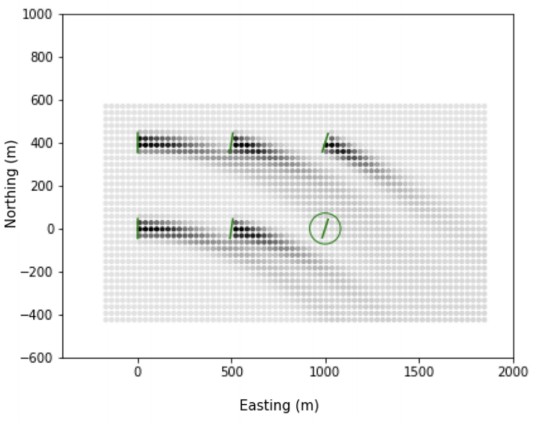
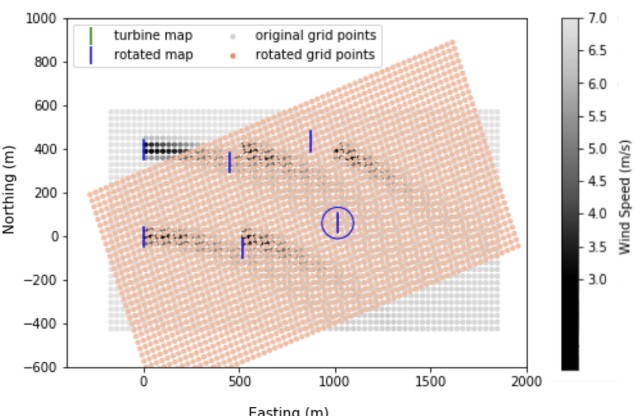

(a) Non-rotated (original) flow-field grid points, showing all but one of the turbine wakes calculated.

(b) The flow-field grid points are rotated about the center of rotation using the wind direction defined at the lower right turbine (20°), and translated to be aligned with the location of the turbine.

**Figure 6.** Depiction of the process performed in FLORIS to align the flow-field grid with the location and wind direction of turbine T6 (as defined in Fig. 5a).

Next, to calculate the velocity deficit caused by each turbine's wake, all of the grid points in the flow field are rotated to replicate the effects of changing wind direction. These rotated grid points represent the redirection of the flow in response to changing wind direction within the flow field (see Fig. 7a). Once the velocity deficit has been calculated using the rotated grid points, the points are rotated back to their original positions in the flow field. Fig. 7b shows the product of the final step, where the calculated velocity deficit is subtracted from the initial free-stream velocity at each flow-field grid point to reveal the resulting shape of the wake.

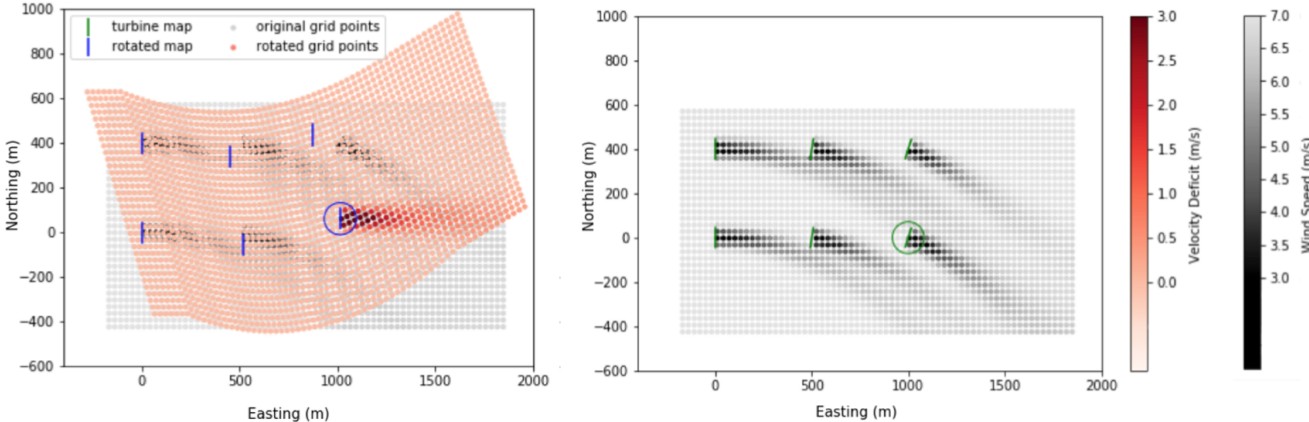

(a) Rotated grid points used in FLORIS to calculate the velocity deficit behind a turbine.

(b) Non-rotated (original) flow-field grid points showing resulting wake calculations.

**Figure 7.** Visualizations of FLORIS calculating velocity deficit turbine T6 in conditions of heterogeneous wind direction. The velocity deficit is calculated using the grid points in the fully rotated position (Fig. 7a), and then applied to the free-stream velocity defined at the grid points in their original non-rotated location (Fig. 7b).

As discussed in Section 2.2, there is a minor computational expense in simulating the flow field independently for each turbine in the wind farm. This is because FLORIS determines a unique set of rotated grid points relative to the wind direction and coordinates of each turbine separately. The grid spacing in the streamwise ($x$) direction relative to the direction of flow is kept uniform throughout each iteration of the rotated grid, but the spanwise ($y$) spacing is adjusted with respect to the local wind direction inside the flow field. This allows the model to replicate a gradual change in wind direction throughout the flow field. The resulting flow-field wake calculation is shown in Fig. 8.

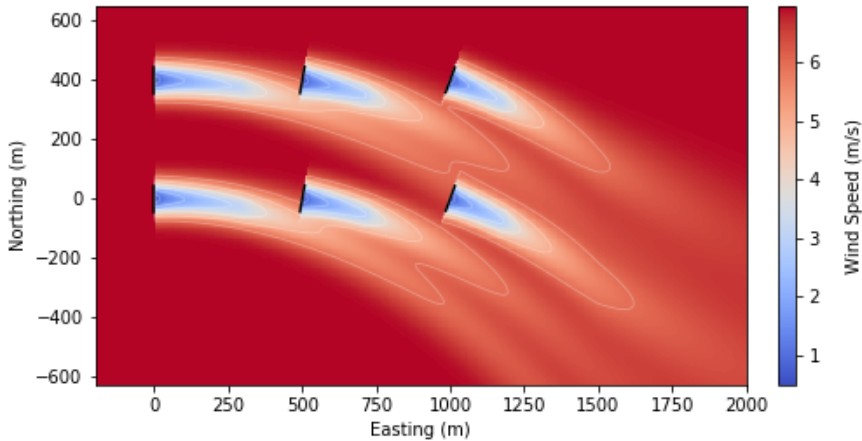

**Figure 8.** Visualization of a flow field with heterogeneous wind direction. Turbine rotors are indicated by black lines.

The grid point spacing in the $x$ direction must be kept constant to avoid elongation or distortion of wake propagation and placement. Because the grid spacing in the $y$ direction is not kept uniform, it must be noted that this capability of emulating a gradual change in wind direction may prevent the model from conserving momentum in some situations. Methods of enforcing uniform spacing in the $y$ direction for each individual turbine wake have been developed, but are not currently implemented because doing so limits the model's ability to create a gradient of wind directions within the flow field. In future work, methods of enforcing momentum conservation in this algorithm will be further investigated.

To further exemplify the applications of this functionality, Fig. 9 shows a more complex simulation of non-constant heterogeneous wind direction simulation in an irregularly spaced wind farm. The steps that FLORIS performs to evaluate this flow condition are identical to the ones displayed in Figs. 5 - 7, except it is personalized to the more complex variations of the depicted state of flow.

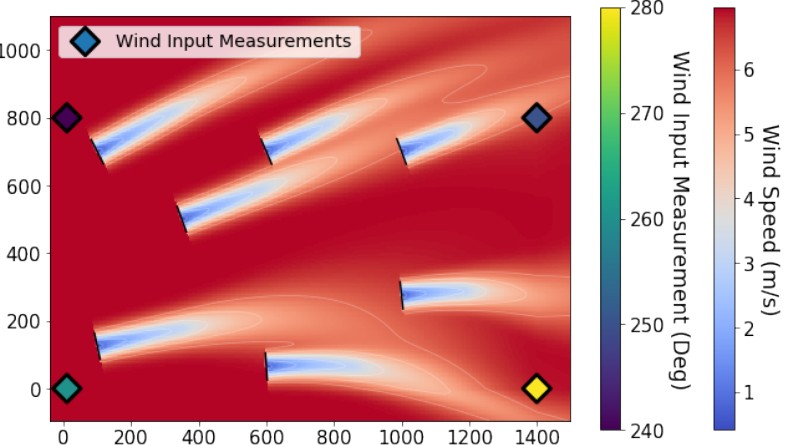

**Figure 9.** A second visualization of a flow field with more complex heterogeneous wind direction. Wind input measurements are indicated using diamond markers, and turbine rotors are shown with black lines.

It is important to consider that this model was not designed to calculate the effects of changes in wind direction that are extremely dynamic. A change in wind direction that is too drastic will cause grid points in the rotated flow-field grid (the red points shown in Fig. 7a) to overlap each other within the same coordinate system, which may result in erroneous assignment of velocity deficit to these overlapped points. The limiting amount of wind direction change for the heterogeneous model is therefore the amount that causes the flow-field grid points to overlap in this manner. This limit must be determined for each wind farm independently, since it is dependent on the site-specific layout geometry and wind conditions of each case.

Although it may be possible for the wind direction within a wind farm to change this drastically, these conditions often involve multiple adjacent domains of flow that are separated by a boundary, which are difficult to represent in this model. These weather conditions are also most often observed in instances of lower wind speeds, and therefore can be considered not as lucrative in regards to power production. Plans for future developments to FLORIS involve designing a more inclusive model that is capable of mitigating issues concerning rapid changes in wind direction.

### 3.4 Heterogeneous turbulence intensity

The geographic distribution of turbulence intensity is established for the initial state of the flow field through the interpolation methods discussed in section 3.1. This strategy of defining a more detailed variation of turbulence intensity in the flow field makes approximation of wake dissipation and deflection more accurate, therefore improving the estimation of the effect of nearby turbine operation within a wind farm. The implementation of heterogeneous turbulence intensity and heterogeneous wind speed are similar, in that the initial heterogeneous conditions are established throughout the flow field by interpolating from the input values, and then waked conditions are updated throughout FLORIS computations of flow-field interactions. During the calculation of wakes, the ambient turbulence intensity that is initially defined at each turbine location is continuously recalculated to account for added turbulence intensity resulting from turbine wakes up to $15D$ upstream, as previously discussed

in Section 2.3.2 and in Niayifar and Porté-Agel (2015). A horizontal plane of a FLORIS simulation featuring heterogeneous turbulence intensity can be observed in Fig. 10.

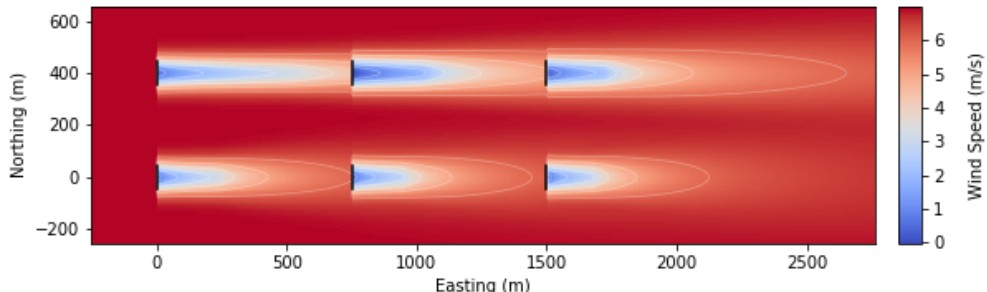

**Figure 10.** Visualization of a flow field with heterogeneous turbulence intensity. The turbines that experience higher turbulence intensity show a faster rate of wake recovery, and vice versa. Turbine rotors are indicated by black lines.

It is important to note that in the interest of conserving computational efficiency, calculations for evaluating the rate of wake expansion and recovery are only dependent on the updated turbulence intensity at the location of the turbine creating the wake.

### 3.5 Turbulence correction

In addition to the heterogeneous features, developments were also made to reduce inaccuracies in power-output predictions caused by turbulent operating conditions. As mentioned in Section 2.1, the accuracy of the zero-turbulence power curve is compromised in conditions of varying turbulence intensity. The revised power calculation, presented in this section, includes a parameter that approximates the effect of turbulence intensity on the power output of a turbine in a wind farm.

Specifically, this approach adjusts the power output with respect to the level of turbulence intensity at a turbine. The adjusted power is calculated by using distribution of the wind-speed fluctuations at the turbine, based on calculations that consider the original wind speed and the standard deviation in wind speed. The first step in this algorithm is to create a normalized probability density function, $f(x)$, of wind speeds, $x$, evenly distributed within the domain of one standard deviation from the mean wind speed, $\mu$. The standard deviation, $\sigma$, is determined by multiplying the turbulence intensity at the turbine by the mean wind speed, $\mu$. Wind speeds that are greater than the cutout wind speed are omitted.

The value of the power coefficient, $C_P$, in the power table is also determined at each wind speed, $x_i$, and at the original wind speed ($\mu$). The ratio of the adjusted power ($P_{adj}$) to the original value of power ($P_0$) is referred to as the turbulence parameter, $\Lambda$. The turbulence parameter can be calculated by summing the weighted adjusted values of power in the following expression, for each wind speed, $x_i$, in the domain of the probability density function, $f(x_i)$:

$$\Lambda = \frac{P_{adj}}{P_0} = \frac{\int_{x_1}^{x_{100}} f(x_i, \mu, \sigma) C_{P,i} x_i^3 dx_i}{C_{P,\mu} \mu^3} = \frac{\sum_{i=1}^{100} f(x_i, \mu, \sigma) C_{P,i} x_i^3}{C_{P,\mu} \mu^3}, \tag{12}$$

where the integral of $f(x_i)$ is approximated by taking 100 samples of the $f(x_i)$. The resulting power curves depending on turbulence intensity are shown in Fig. 11. As the turbulence intensity increases, the power output increases in Region 2 and decreases across Region 3.

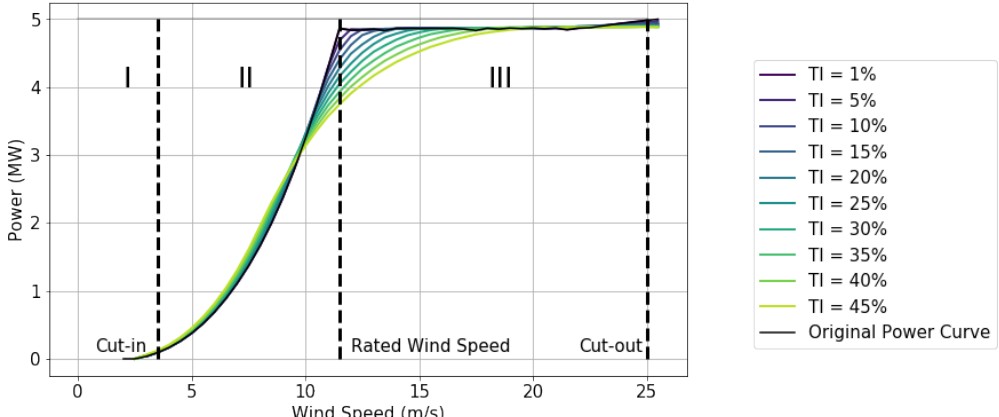

**Figure 11.** Adjusted power curve for the NREL 5-MW reference turbine for different turbulence intensities. The dashed lines denote the cut-in, rated, and cut-out wind speeds, and also represent the boundaries of the first, second, and third regions, respectively.

The following expression may be used to calculate the final value of adjusted power output, $P_{adj}$, with respect to the current turbulence intensity at a turbine:

$$P_{adj} = P_0 \Lambda = \frac{1}{2} \rho A C_{P,\mu} \cos^p(\gamma) \mu^3 \Lambda. \tag{13}$$

Where $\gamma$ is the yaw angle of the turbine, and $\Lambda$ represents the turbulence parameter. The value of $\Lambda$ must always be greater than zero.

In future work, this turbulence-correction model could be improved by implementing a similar consideration of the thrust coefficient, $C_T$. Because the velocity deficit computations in this model rely on the value of $C_T$, it may be advantageous to expand this method to calculate an adjustment parameter for the effects of turbulence on rotor thrust.

It is important to note that similar models have been developed that incorporate methods of turbulence re-normalization based on machine-learned or empirically-derived data (Clifton and Wagner, 2014). The proposed method discussed in this section was developed to attempt to represent the variation of power output due to turbulence effects, while using a simple strategy that is not dependent on the availability of data other than the current wind farm atmospheric measurements, and the power curve provided by the turbine manufacturer. In future work, it may be advantageous to incorporate more complex techniques that are able to capture the effects of turbulence intensity with greater detail and accuracy.

## 4 Model validation and analysis

Two validation studies are presented to analyze the effectiveness of the adapted FLORIS wake model. In Section 4.1, a 38-turbine wind farm is simulated using the heterogeneous FLORIS model and compared to results from large eddy simulations to evaluate the accuracy of wind farm power predictions. Additionally, Section 4.2 presents a study where a large wind farm is simulated using the heterogeneous FLORIS model and turbulence power calculations. The results of these FLORIS simulations are compared to the wind farm's SCADA data records to further evaluate the model's power prediction accuracy and wake model performance.

### 4.1 Comparisons to LES

This section presents a validation study that evaluates the power prediction accuracy of the proposed heterogeneous FLORIS model in comparison to large eddy simulations of a 38-turbine array, calculated using NREL's tool Simulator fOr Wind Farm Applications (SOWFA) (Fleming et al., 2013). The simulated wind farm contains 38 turbines modeled with NREL's 5MW reference turbine design criteria (Jonkman et al., 2009), and arranged in a concentric circular layout similar to Thomas et al. (2019); Fleming et al. (2020a); King et al. (2020a).

Twelve test cases were evaluated for this study; each were simulated using different wind directions, varying from 10 degrees to 340 degrees in 30 degree intervals. Spatially homogeneous inputs were used to simulate wind direction and turbulence intensity, where turbulence intensity was at 9% for all cases. The free-stream wind speed remained close to 8 m/s for all cases, with minor spatial variations. FLORIS wind speed inputs were obtained by extracting the free-stream velocity from LES results at locations upwind of turbines in undisturbed flow. These extracted input values create a velocity gradient in the direction normal to the wind direction in the heterogeneous FLORIS model, as seen on the right in Fig. 12. The wind speed input for the homogeneous FLORIS model was obtained by taking an average of the heterogeneous input values for each case.

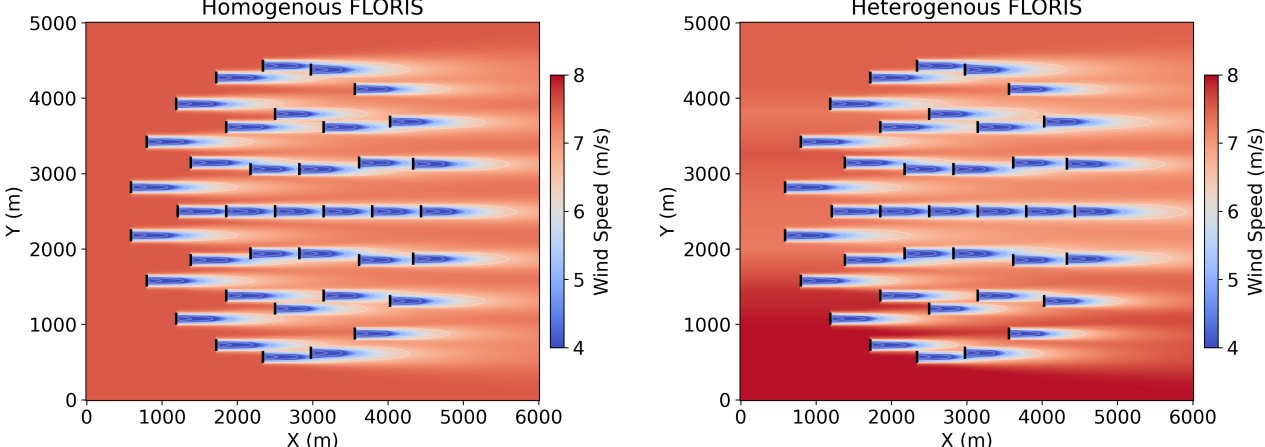

**Figure 12.** Horizontal cut-planes of the FLORIS simulations with wind direction at 270 degrees. Wake losses were calculated using the homogeneous FLORIS model (left) and heterogeneous FLORIS model (right). Turbine rotors are indicated by black lines.

To analyze the effects of wake losses for the simulated wind farm, additional heterogeneous and homogeneous FLORIS simulations were conducted excluding all FLORIS wake calculations. Fig. 13 shows the total wind farm output predictions from all four FLORIS simulation models, and compares them to the LES case result (shown in black). The trends observed Fig. 13 indicate that the models which neglected wake loss calculations drastically over-estimated total wind farm power output, with the heterogeneous model reporting more accurate power predictions overall. Alternatively, the FLORIS models that did perform wake loss calculations produced a much more accurate estimation of wind farm power output.

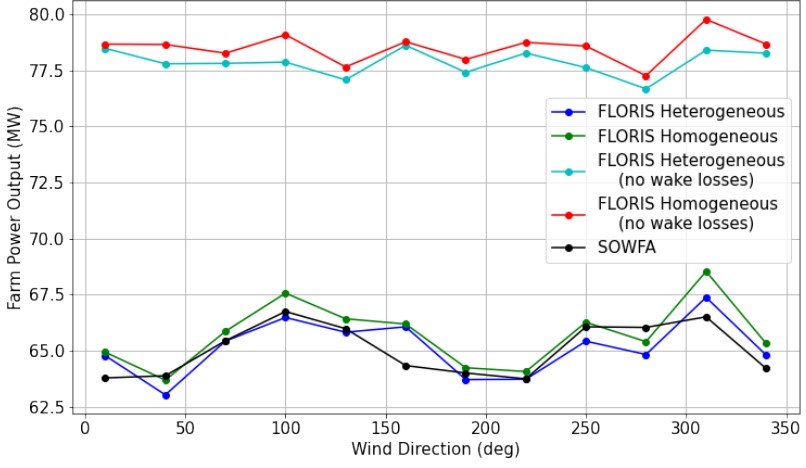

**Figure 13.** Total wind farm power output calculated using the homogeneous and heterogeneous FLORIS models, with and without wake losses. The results of the SOWFA simulation are also plotted for reference in black.

The absolute error of the total wind farm power output was calculated for each model to analyze power prediction accuracy at every wind direction evaluated in the case study. The results of these calculations are shown in Fig. 14, and each model's average absolute error over all wind directions is recorded in Table B1 for reference. In Table B1, an 18.9% decrease in average absolute error is reported when using the heterogeneous wake model compared to the homogeneous wake model.

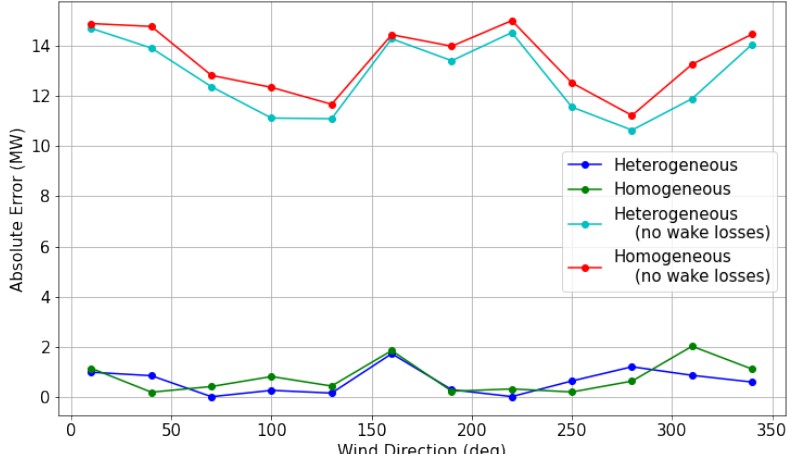

**Figure 14.** Absolute error in total farm power output calculated using the homogeneous and heterogeneous FLORIS models, with and without wake losses.

The Mean Absolute Error (MAE) of power predictions at individual turbines in the wind farm was also calculated using Equation 14.

$$MAE = \frac{1}{n} \sum_{i=1}^{n} |P_{model,i} - P_{actual,i}| \tag{14}$$

where $n$ is the number of turbines in the wind farm, $P_{actual,i}$ is the LES-generated wind farm power output for turbine $i$, and $P_{model,i}$ represents the predicted power output from the FLORIS model for turbine $i$. MAE was calculated for each FLORIS model at every wind direction in the case study. The resulting MAE values are shown in Fig. 15 and the average MAE for all wind directions is reported in Table B2 for each FLORIS model.

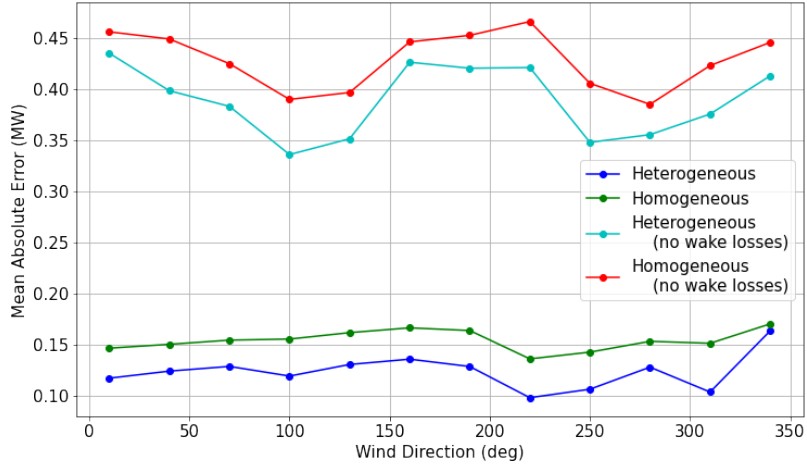

**Figure 15.** Mean absolute error of individual turbine power output predictions calculated using the homogeneous and heterogeneous FLORIS models, with and without wake losses.

In comparison to Fig. 14, Fig. 15 indicates a more observable and consistent disparity in power output accuracy between the results of heterogeneous and homogeneous FLORIS wake simulations. Table B2 validates this observation by reporting a 385 19.5% decrease in MAE at each turbine when using the heterogeneous model. The data provided in this comparison confirms that the proposed heterogeneous model offers substantial advancements in the generation of accurate power predictions at individual turbines within a wind farm.

## 4.2 Comparisons to wind farm SCADA data

This section summarizes a validation study presenting comparisons of FLORIS power predictions to SCADA-recorded power 390 outputs from an observed operational wind farm. A large, utility-scale wind farm located within mountainous terrain was chosen for this study because it is often subject to unpredictable and dramatic shifts in weather conditions. More information regarding the physical layout and characteristics of this wind farm can be found in Appendix A. The motivation behind performing these simulations was to quantify the effect of the recent developments to FLORIS in reducing the error in power-output predictions for wind farms in complex terrain.

FLORIS simulations were performed using heterogeneous inputs of wind direction, turbulence intensity, and wind speed, which were taken from the wind farm's SCADA records. These inputs include four wind measurement values for each atmospheric characteristic, derived from Meteorological (MET) tower measurements placed in various locations throughout the wind farm. Similar simulations were performed using an identical FLORIS model, but with a singular homogeneous input for wind speed, wind direction, and turbulence intensity. These homogeneous inputs were derived by evaluating the average of the 400 five heterogeneous input values at each time step. The resulting power output of all simulations was recorded with the inclusion

of the turbulence correction and without. All cases were simulated using data averaged at time steps of 10 minutes over a range of 2 months.

In the following discussion, the results from all FLORIS simulations are presented and analyzed to determine the accuracy of power predictions from each test case. Fig. 16 includes two horizontal planes showing a partial section of heterogeneous flow calculations during these simulations. This figure demonstrates the visual capabilities of the heterogeneous model and how the effects of the new wake calculations can be translated into visual information for further analysis of wake interactions within a wind farm.

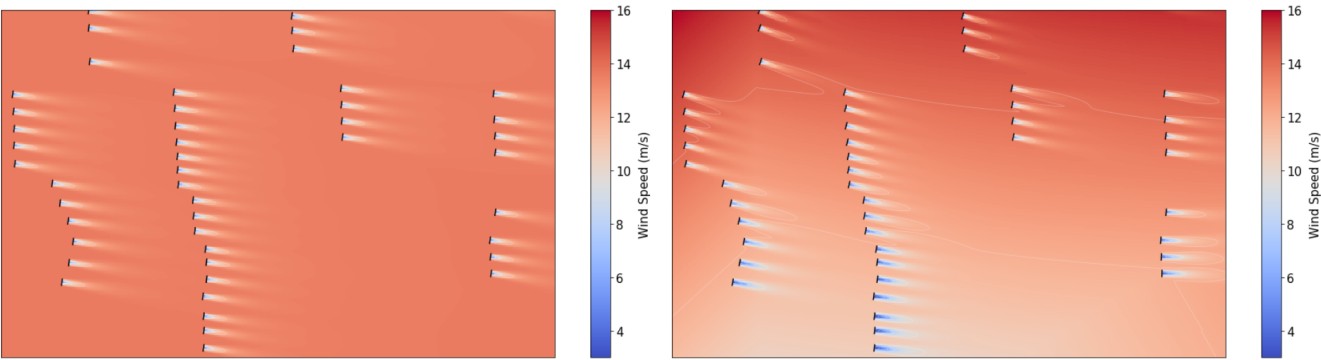

(a) Horizontal plane of a FLORIS simulation using the homogeneous model. (b) Horizontal plane of a FLORIS simulation using the heterogeneous model.

**Figure 16.** Horizontal planes of two different FLORIS simulations, taken at the same time-step iteration.

Although these visualizations do not give direct estimates of power prediction, they are helpful in translating the input measurements into a form that characterizes the general behavior of wind farm dynamics for the interpretation of the observer. The cut plane visualization is helpful in performing qualitative analysis of turbine wake interactions, and is more useful when displaying the estimated weather conditions characteristic of each location in the flow field, which is improved in the heterogeneous model.

When comparing the performance of the simulations, the calculated power output was tabulated and compared, for accuracy. In Fig. 17, the sum of wind farm power output from each FLORIS simulation is normalized with respect to the rated power output for the wind farm, and plotted along with the recorded SCADA output. This approach highlights any weaknesses in each model, relative to the overall performance of the others. A 24-hour period was chosen to demonstrate how the models performed under average diurnal conditions. Figure 17 shows a day with relatively variant weather conditions and many rapid shifts in power output.

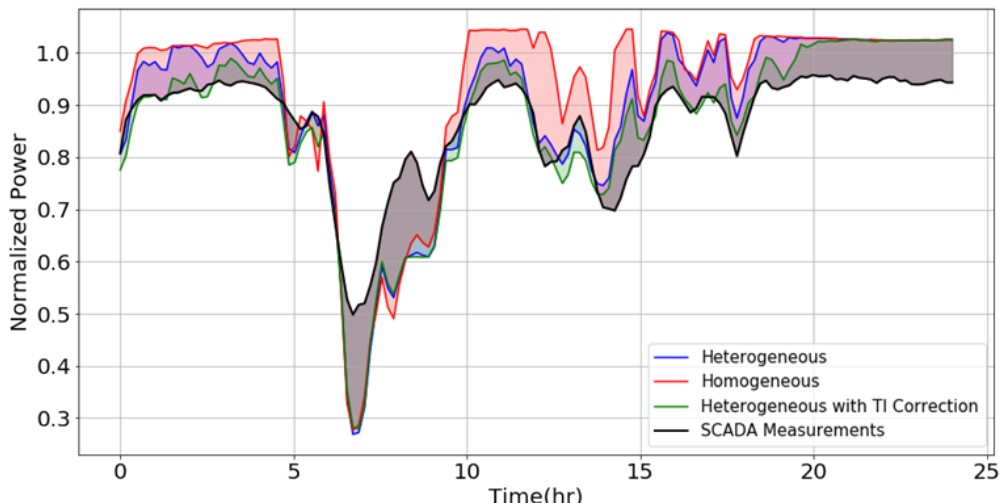

**Figure 17.** Power output calculated by FLORIS for homogeneous (red), heterogeneous without the turbulence correction (blue), and heterogeneous with the turbulence correction (green), compared with SCADA data shown in black. Each shaded region represents the difference between predictions of power output, and the measured power output from SCADA data.

In Figure 17, it is evident that the heterogeneous models are predicting the power output more accurately than the homogeneous model. The trend line of the heterogeneous simulations consistently follows closer to the line representing the power output recorded from SCADA data. Additionally, the heterogeneous simulation that included turbulence-intensity corrections showed an extra advantage in estimating turbine performance, following closely to the trend line of the heterogeneous simulation, and also reliably contributing error-reducing improvements to the heterogeneous model. While this juxtaposition is effective in ranking each model's ability to estimate total farm power output, it should be noted this comparison only indicates the accuracy of a calculations for the entire wind farm power output collectively, without considering the accuracy at each turbine individually.

It is possible for wake models to overpredict the power output of some turbines, and underpredict others, in a way that produces a total wind farm power estimate that seems accurate, but is not using reliable and precise methods of calculation. To verify that the recent additions to FLORIS have improved the power-predicting capabilities, it must be confirmed that the new model produces a consistently accurate estimate with respect to each iteration in the time series and each turbine within the wind farm individually. To prove this model's consistency in accuracy, the normalized absolute error was calculated at each turbine at each iteration of the time series for this same day. The sum of the absolute error at all turbines within the wind farm is calculated for each simulation model at each time iteration. To calculate the sum of absolute error (SAE) for all turbines, the following formula was applied to each time iteration of the simulation.

$$SAE = \sum_{i=1}^{n} |P_{model,i} - P_{actual,i}|, \tag{15}$$

where $n$ is the number of turbines in the wind farm, $P_{actual,i}$ is the measured power output of turbine $i$, and $P_{model,i}$ is the predicted power output of turbine $i$ from a given FLORIS model. The results of each FLORIS model were calculated and plotted on the same set of axes in Figure 18.

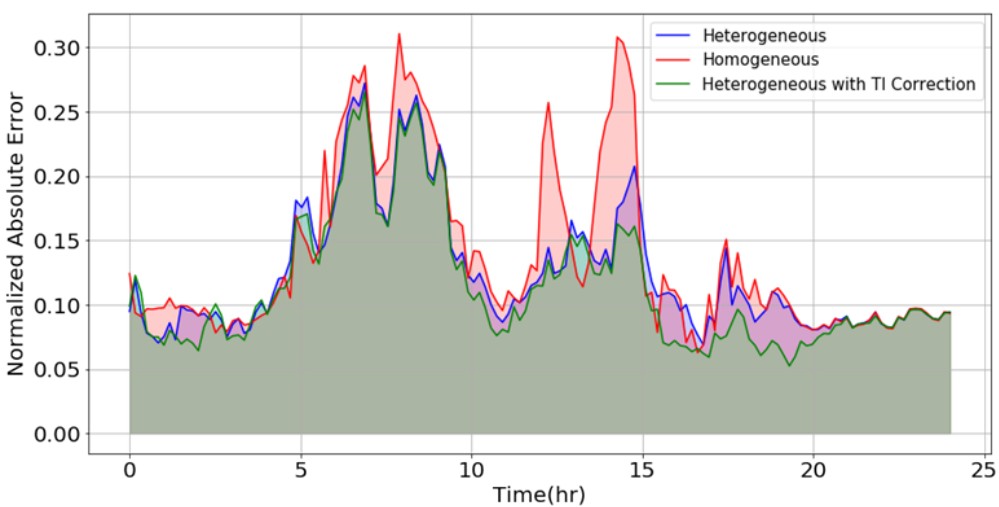

**Figure 18.** Sum of the normalized absolute error at each turbine in the wind farm, computed at each time step.

The trends observed in Fig. 18 exhibit similar characteristics that indicate the accuracy of the model at each turbine is increasing with the application of the heterogeneous model and turbulence-intensity correction parameter. The heterogeneous model reliably produces less error when calculating the power at each turbine over the time series, which ensures that the power predictions of the entire farm are not self-compensating because of simultaneous overpredictions and underpredictions of individual turbine outputs. Furthermore, if Figure 18 is analyzed with respect to the trends of normalized power in Figure 17, it is evident that the addition of heterogeneity and turbulence-intensity corrections contributes to improving the accuracy of FLORIS power predictions in instances of overprediction and underprediction, and transitions between the two with relative consistency.

To ensure these same trends of accuracy persist over the entire two-month period, the percent error of the total wind farm power output was calculated at each time-step iteration using the following equation.

$$Percent\ Error = \frac{|P_{model} - P_{actual}|}{|P_{actual}|}, \tag{16}$$

where $P_{actual}$ is the measured power output of the wind farm, and $P_{model}$ is the power output of the wind farm predicted by a given FLORIS model. The results of these calculations were grouped into three separate domains: wind speeds of less than 5 m/s, wind speeds in the range of 5 to 11 m/s, and wind speeds greater than 11 m/s. Time iterations when wind speed was less than the cut-in wind speed (2.5 m/s) were considered negligent in regards to power production and therefore omitted from the

data set. A histogram of the percent error in each wind-speed domain was computed over the entire time series to display the distribution of error with respect to each simulation (Fig. 19).

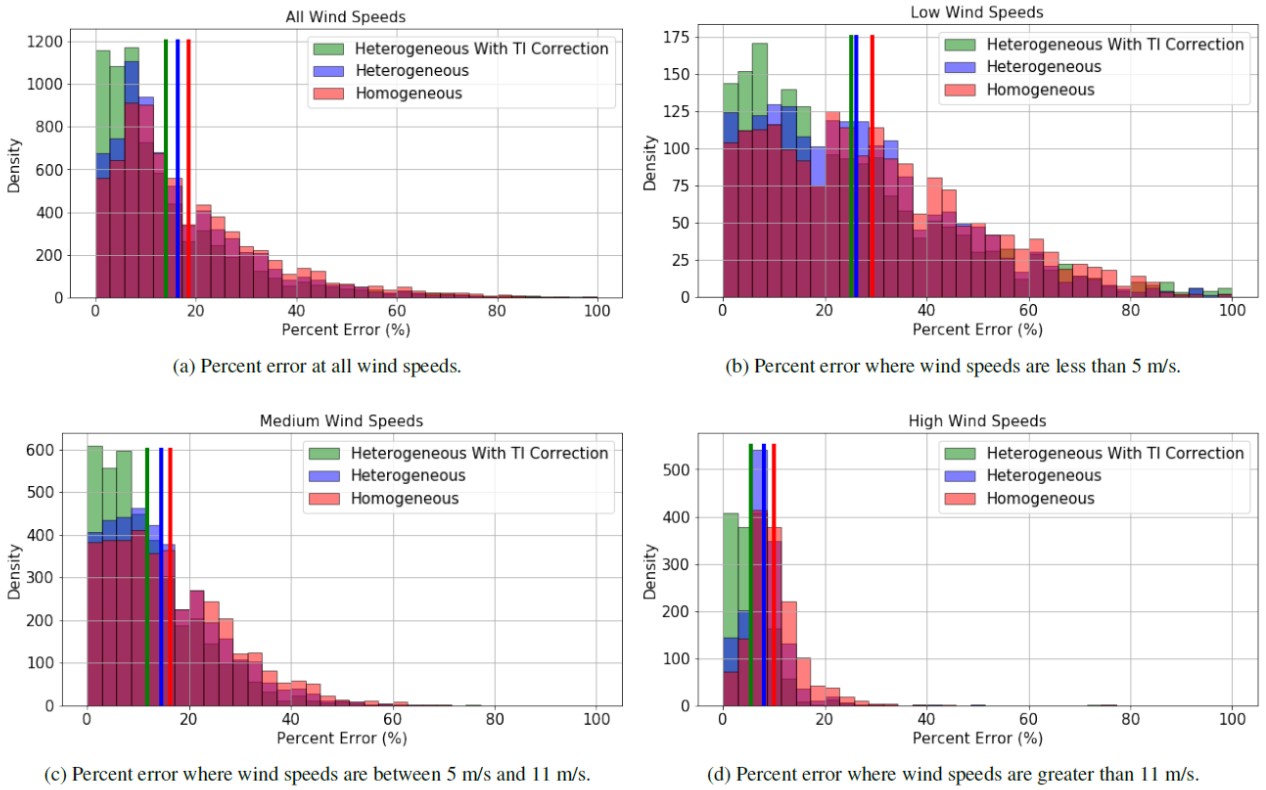

(a) Percent error at all wind speeds.

(b) Percent error where wind speeds are less than 5 m/s.

(c) Percent error where wind speeds are between 5 m/s and 11 m/s.

(d) Percent error where wind speeds are greater than 11 m/s.

**Figure 19.** Percent error of all three FLORIS models, plotted for comparison within varying ranges of wind speeds.

Although the plots for the wind-speed domains vary slightly in distribution, it is clear that each histogram exemplifies a trend toward accuracy in simulations that incorporate heterogeneity and turbulence-correction calculations. It is important to note that only the data points shown in the percent-error range of each histogram were used to calculate the respective binned averages. The outliers were omitted because they tend to skew the presentation of the data set in a way that obscures the actual trend of data.

The mean absolute percent error (MAPE) of all time-step iterations are also reported in Table 1. The data for this table was calculated by evaluating the percent error of FLORIS power predictions for the full wind farm at each time step, and then solving for the mean over the entire time series. This calculation is expressed as:

$$MAPE = \frac{1}{n} \sum_{i=1}^{n} \frac{|P_{model,i} - P_{actual,i}|}{|P_{actual,i}|}, \tag{17}$$

where $n$ is the number of time steps in the total simulation, $P_{actual,i}$ is the recorded power output of the wind farm at time step $i$, and $P_{model,i}$ denotes the predicted power output from the FLORIS model at time step $i$.

**Table 1.** Mean absolute percent error in total wind farm power output for all FLORIS models, tabulated for comparison within varying ranges of wind speeds.

| FLORIS Simulation Model | Mean Absolute Percent Error at Wind Speed (%) | | | |
| --- | --- | --- | --- | --- |
| | < 5 m/s | 5 - 11 m/s | > 11 m/s | all |
| Homogeneous | 43.2 | 16.3 | 10.0 | 22.4 |
| Heterogeneous | 48.2 | 14.5 | 8.0 | 22.5 |
| Heterogeneous with Turbulence-Intensity Correction | 61.4 | 11.8 | 5.5 | 24.2 |

When comparing the MAPE values in Table 1 with the histograms of Fig. 19, an increase in MAPE is observed in Table 1 for lower wind speeds of simulations that implemented heterogeneous and turbulence correction models. This is a trend that is not characteristic of the histograms depicted in Fig. 19b. In reference to this observation, is important to note that the metric of MAPE penalizes overpredictions with more weight than underpredictions. Furthermore, MAPE calculates mean with equal weight for all time steps in the data set, which is not always ideal for an indication of overall farm power output accuracy. It is possible that the reported increase in MAPE with lower wind speeds may be an indication that the heterogeneous and turbulence intensity correction models tend to produce more frequent overpredictions of power output in conditions where wind speeds are near the cut-in speed. If this is true, further investigations may be conducted in future work to determine why this is happening and how it could be circumvented.

Although MAPE is an informative metric for analyzing the average percent error relative to a specific power output range, methods that use unweighted averaging are sometimes misleading in the analysis of overall power prediction accuracy. The relative error during time-step iterations with lower power output can seem large, even when the absolute error is insignificant in comparison to the magnitude of total farm output.

A more comprehensive representation of relative model accuracy is presented in the following table, where the mean absolute error (MAE) is evaluated for total wind farm output. This was calculated by evaluating the absolute error at each time step, and then taking the mean of these error values. This calculation is performed using Eq. 14, where $n$ is the total number of time steps in the simulation, $P_{actual,i}$ is the recorded power output of the wind farm at time step $i$, and $P_{model,i}$ is the predicted power output from the FLORIS model at time step $i$.

By taking an average of absolute errors instead of relative errors, MAE is a more effective metric in representing the overall accuracy of total wind farm power prediction. The resulting MAE values are shown in Table 2, where a clear trend of increased accuracy is observed for models that implement heterogeneity and turbulence-adjustment calculations, including the cases where wind speeds are below 5 meters per second.

**Table 2.** Mean absolute error in total wind farm power output for all FLORIS models, tabulated for comparison within varying ranges of wind speeds. Total rated wind farm output was scaled to 100 MW for reference.

| FLORIS Simulation Model | Mean Absolute Error at Wind Speed (MW) | | | |
|---|---|---|---|---|
| | < 5 m/s | 5 - 11 m/s | > 11 m/s | all |
| Homogeneous | 4.7 | 25.7 | 38.7 | 22.6 |
| Heterogeneous | 4.2 | 22.8 | 31.4 | 19.4 |
| Heterogeneous with Turbulence-Intensity Correction | 4.1 | 19.0 | 22.0 | 15.5 |

Lastly, values of MAE were also calculated to represent the accuracy of the model at each individual turbine within the wind farm. Using this metric, Table 3 shows that simulations implementing the heterogeneous model and turbulence correction calculations outperformed the homogeneous model in the prediction of individual turbine power output. This should be expected, since the overall farm output in Table 2 followed a similar trend. The marked improvement of power predictions at individual turbines suggests that the addition of the proposed heterogeneous and turbulence correction methods enhance the FLORIS wake model by simulating farm-flow interactions with more thorough detail and greater accuracy.

**Table 3.** Mean absolute error in individual turbine power output for all FLORIS models, tabulated for comparison within varying ranges of wind speeds. Total rated wind farm output was scaled to 100 MW for reference.

| FLORIS Simulation Model | Mean Absolute Error at Wind Speed (MW) | | | |
|---|---|---|---|---|
| | < 5 m/s | 5 - 11 m/s | > 11 m/s | all |
| Homogeneous | 0.046 | 0.244 | 0.199 | 0.152 |
| Heterogeneous | 0.041 | 0.208 | 0.191 | 0.133 |
| Heterogeneous with Turbulence-Intensity Correction | 0.041 | 0.202 | 0.179 | 0.129 |

To analyze the influence of wake effects in this study, an identical set of simulations were performed excluding FLORIS wake loss calculations, and the results for MAE at the overall farm and individual turbine levels are reported in Tables B3 and B4 in Appendix B. These simulations seem to indicate that the wake losses at the observed subject wind farm are relatively small, due to the large spacing between turbines in the stream-wise direction. The study presented in Section 4.1 analyzes the performance of the proposed heterogeneous model in a wind farm with more influential wake losses to give a more detailed analysis of the effects of wake losses.

As noted in Section 3.3, the implementation of methods utilized to simulate spatially variant wind direction causes the heterogeneous model to be marginally less efficient in computation. To quantify this increased computational cost, each simulation was timed in this study. On average, these time records show that the simulations using the heterogeneous model took less than 10% longer to compute than those using the homogeneous model. The choice to sacrifice computational efficiency in the heterogeneous model was seen as a necessary trade-off to achieve greater detail and accuracy in simulations of more dynamic environments. Future developments to FLORIS will attempt to optimize the efficiency of this model, and reduce the time necessary to simulate the effects of changing wind direction.

## 5 Conclusions

This article introduces a method to include heterogeneous flow fields into the FLORIS simulation tool, as well as a turbulence correction to the power reported at each turbine. To analyze the developed model's improvements in accuracy, several FLORIS simulations with and without these changes were compared to large eddy simulations and SCADA data from a utility-scale wind farm. The results of the FLORIS simulations indicate that these two modifications improve power predictions of the wind farm at the turbine and wind farm level. The increased accuracy of this model's power-prediction capabilities shows that this method is more precise in predicting farm-flow interaction in heterogeneous and turbulent environments, which previous versions of FLORIS were not able to simulate.

Overall, the heterogeneous and turbulence-intensity correction modifications presented in this article showed a positive effect on the accuracy of FLORIS capabilities. This improved model provides a more detailed quantitative and qualitative analysis of wind farm flow, including the demonstration of heterogeneous flow in cut-plane velocity plots, and improved accuracy in power prediction at individual turbines as well as total wind farm power output. Comparing FLORIS power predictions to LES, the heterogeneous FLORIS model showed an 18.9% decrease in Mean Absolute Error (MAE) for total wind farm power output, and a 19.5% decrease in MAE for individual turbine power predictions compared to the homogeneous FLORIS model. In comparisons to SCADA data, FLORIS simulations that implemented the heterogeneous flow model showed a 14.6% decrease MAE for wind farm power output predictions compared to homogeneous model simulations. With the use of the proposed Turbulence Intensity Correction method in addition to the heterogeneous model, the MAE in farm power output predictions showed a 31.42% MAE decrease compared to the homogeneous model.

These modifications to FLORIS have outlined a framework for a wake model that features atmospheric heterogeneity and turbulence-intensity corrections to the power curve, and provides a platform for further developments in this area of research. In agreement with this study, the findings of Fleming et al. (2020a) also indicate that this model shows promise in enhancing the performance of FLORIS's existing wind farm optimization controls, in addition to improving the accuracy of wind farm power predictions.

Further studies relating to the effectiveness of this model when applied to wind farm controls would be very beneficial in determining future developments to these algorithms. Additionally, more extensive investigations should be considered to evaluate the efficacy of the proposed model in a wider variety of operational conditions, particularly those with lower wind speeds and extreme variations in wind direction. Other future work will investigate alternative interpolation methods for the flow-field that consider the wind farm terrain map, capabilities for simulating more dynamic changes in wind direction, implementing enforcement of momentum conservation, and optimizing the model's computational efficiency.

**Appendix A: Wind Farm characterization**

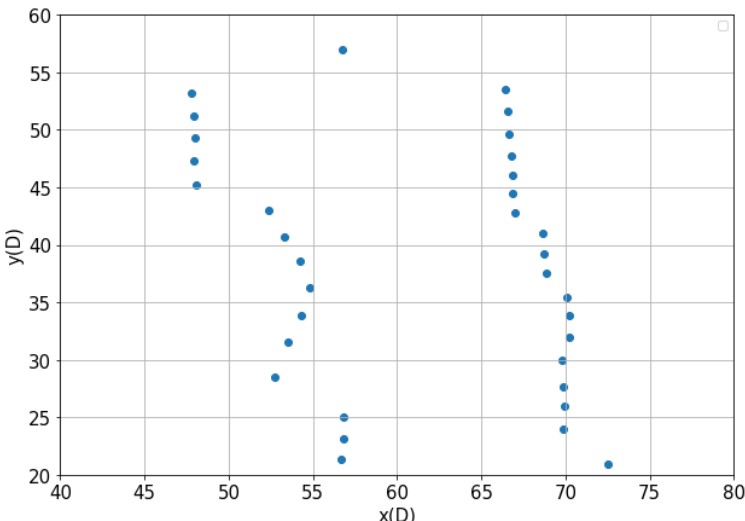

**Figure A1.** Map of a selected section of the observed wind farm discussed in Section 4.2. This plot show the inter-distance between turbine locations in the Northing (y) and Easting (x) directions. The distances shown on each axis are labeled relative to the average rotor diameter (D) of the turbines in the wind farm.

**Table A1.** This table lists several key attributes that characterize the nature of the terrain and turbine layout of the observed wind farm discussed in Section 4.2. Distance values are reported relative to the average turbine rotor diameter (D). Span-wise and stream-wise directions are defined to be perpendicular and parallel to the average wind direction during the wind farm, respectively.

| Measured Quantity | Distance in terms of average rotor diameter (D) |
| --- | --- |
| Average stream-wise inter-distance | 20.0 D |
| Average span-wise inter-distance | 2.0 D |
| Range of elevation variation | 2.2 D |

## Appendix B: Additional Results

**Table B1.** Absolute error of total wind farm power output predictions for four different FLORIS models compared to LES simulations, as discussed in Section 4.1. An average absolute error is reported, which was taken over all wind direction cases.

| FLORIS Simulation Model | Average Absolute Error of Wind Farm Power Output (MW) |
|---|---|
| Heterogeneous | 0.636 |
| Homogeneous | 0.784 |
| Heterogeneous without wake losses | 12.796 |
| Homogeneous without wake losses | 13.450 |

**Table B2.** Mean absolute error of individual turbine power outputs for four different FLORIS models compared to LES simulations, as discussed in Section 4.1. An average MAE is reported, which was taken over all wind direction cases.

| FLORIS Simulation Model | Average MAE of Individual Turbine Power Output (MW) |
|---|---|
| Heterogeneous | 0.124 |
| Homogeneous | 0.154 |
| Heterogeneous without wake losses | 0.389 |
| Homogeneous without wake losses | 0.428 |

**Table B3.** Analysis of wake influence in the observed wind farm discussed in Section 4.2. This table shows the average mean absolute error in total wind farm power output for three different FLORIS models, omitting FLORIS wake loss calculations. This average was taken over all time steps, and the total rated wind farm output was scaled to 100 MW for reference.

| FLORIS Simulation Model | MAE for Overall Wind Farm Power Output (MW) | | | |
|---|---|---|---|---|
| | < 5 m/s | 5 - 11 m/s | > 11 m/s | all |
| Homogeneous | 4.9 | 26.1 | 38.2 | 22.7 |
| Heterogeneous | 4.6 | 28.1 | 13.0 | 18.6 |
| Heterogeneous with Turbulence-Intensity Correction | 4.0 | 24.9 | 21.9 | 18.5 |

**Table B4.** Analysis of wake influence in the observed wind farm discussed in Section 4.2. This table shows the average mean absolute error of individual turbine power output for three different FLORIS models, omitting FLORIS wake loss calculations. This average was taken over all time steps, and the total rated wind farm output was scaled to 100 MW for reference.

| FLORIS Simulation Model | Average MAE of Individual Turbine Power Outputs (MW) | | | |
|---|---|---|---|---|
| | < 5 m/s | 5 - 11 m/s | > 11 m/s | all |
| Homogeneous | 0.045 | 0.244 | 0.198 | 0.152 |
| Heterogeneous | 0.0415 | 0.229 | 0.263 | 0.155 |
| Heterogeneous with Turbulence-Intensity Correction | 0.042 | 0.223 | 0.263 | 0.152 |

*Acknowledgements.* This work was authored by the National Renewable Energy Laboratory, operated by Alliance for Sustainable Energy,

LLC, for the U.S. Department of Energy (DOE) under Contract No. DE-AC36-08GO28308. Funding provided by the U.S. Department
of Energy Office of Energy Efficiency and Renewable Energy Wind Energy Technologies Office. The views expressed in the article do
not necessarily represent the views of the DOE or the U.S. Government. The U.S. Government retains and the publisher, by accepting the
article for publication, acknowledges that the U.S. Government retains a nonexclusive, paid-up, irrevocable, worldwide license to publish or
reproduce the published form of this work, or allow others to do so, for U.S. Government purposes.

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
