# Peer review of "Design and analysis of a wake model for spatially heterogeneous flow"

_Wind Energy Science, 2020_

## Referee Comment (RC1) · Anonymous Referee #1 · 18 Apr 2020

**General comments**

This paper presents an interesting improvement of the FLORIS wind farm model with the implementation of a method to take into account an heterogeneous atmospheric inflow. The original wind farm model is well described and efforts have been made on the description of the new implementation with plots that are quite useful for the comprehension, but there are still grey areas and it lacks information about the processing of the z-dimension for the complex terrain application: this application is mentioned twice in the introduction and in the conclusion, and the test case is a wind farm in complex terrain, but no information is given on this specific point. While the test case lacks some detailed information about the wind farm and the atmospheric conditions, a comprehensive comparison has been performed between the

original homogeneous FLORIS and the two presented improvements. An exhaustive presentation of quantitative indicators is given and the authors provide well-detailed explanations and conclusions.

Thus, more explanations should be given on the wind direction change processing and the processing of the vertical dimension should also be addressed if a potential application remains "wind farms in complex terrains". Therefore I suggest a major revision.

Here are some general suggestions:

- In the introduction, the authors could mention other state-of-the-art wake modelling utilities and give more general information on the heterogeneity part with reference to studies on the characterization of heterogeneous conditions, the impact of spatial heterogeneity on power predictions...

- About the description of the new implementation, the authors do not mention how they deal with the vertical dimension, especially since they mention in the introduction that a potential application of this new version is wind farms in complex terrains, and the test case is in complex terrains.

- The part addressing the wind direction heterogeneity and the mesh deformation was not very crystal clear for me and needs more details. Maybe a second case without a constant change in wind direction could be interesting.

- About the test case, the authors could give more information about the wind farm (i.e. number and type of turbines, layout/inter-distance), some information about the complexity of the terrain and about the atmospheric measurements at met masts (temporal evolution of wind speed, wind direction and TI for Days A and B, and some information about the stability).

**Specific comments**

- P1, L19: you could mention a reference on the "accurate results in uniform set of atmospheric conditions".

- P3, L68: you mention "yaw-misalignment conditions" but the $\cos(\gamma)$ is missing in Equation 3. However, as you don't consider any yawing strategy in the paper, maybe you could drop the $\cos(\gamma)$ mentions in all equations. Moreover, $u$ should be infinite velocity or without induction zone.

- In Section 2.3 : you could mention the limitations of the Gaussian model (only valid in far wake)

  - P4, L106: you mention a dependence on ambient TI, but it does not appear until Eq 8. You could mention that this dependence is hidden in $k$ with a reference to Eq 8.
  - P4, L114: why do you use quadratic superposition of velocity deficits ? You have an added-TI model and you mention Niayifar and Porté-Agel later: in their paper, they recommend the use of linear superposition of velocity deficit while having an added-TI model.
  - P5, L140: You could nuance this paragraph on turbulence and saturation effect as it is not well understood for now.
  - P6, L145: why is the added-TI model part located in the atmospheric stability section ? Moreover, the equation describing the Crespo model is not correct, it should be $0.73 \times a^{0.8325} \times I_0^{0.0325} \times (\frac{x}{D})^{-0.32}$. You should also mention the validity ranges ($5 < x/D < 15$, $0.07 < I_0 < 0.014$ and $0.1 < a < 0.4$).

- In Section 3:

  - P6, L152: You could specify that the heterogeneous flows are undisturbed atmospheric flows (i.e. without wake effects).

- P6, L158: You could make a reference to Fig 1.a.
- P8, L185: You could name the mentioned algorithms that have been tested for extrapolation, or not mention at all their disadvantages as the explanations are a bit vague and it is difficult to understand what this is about.
- P8, L203: You could mention a reference to Section 3.3. for the processing of wind direction heterogeneity.
- P10, Fig 4 and others: You could name turbines (T1... T6) and make a reference to T6 in the text.
- In general for Section 3.3: this procedure with rotation only works if you have uniform lateral change in wind direction ? Maybe you could choose a more complex case for the wind direction change with a bell behaviour or a S-shape. Moreover, how do you define the centre of rotation ? And how do you deal with wake superposition ?
- P10-11, Fig 5 and 6: You could distinguish rotated grid points for single turbine and rotated grid points for all turbines (you have deformation for this grid).
- In Section 3.4: in this subsection, I can not really say if you deal with heterogeneous ambient/undisturbed TI. It needs some clarification: do you deal with heterogeneous TI the same way you deal with heterogeneous wind speed ? You could add a plot with the corresponding TI in Fig 8.
- In Section 3.5: Have you used an aero-elastic solver for this part ? You could also give an order of magnitude for $\Lambda$.
- P14, L276: You could nuance this comment because having an improved Ct should be as important as having an improved Cp as the velocity deficit model relies on Ct.

• In Section 4:

– P14, L281: More information could be given on the wind farm, the layout (min/max inter-distance), the turbines, the complexity of the terrain...

– P16, Figure 11/12: You could give more information on the daily evolution of wind speed, wind direction and TI, and on $\alpha_s$ to have an information about stability.

• In Section 5:

– P22, L389: You could give an approximate value of the power prediction improvement.

**Technical corrections**

• In general with the plots on wind direction changes, you could add one or two streamlines, it could help in the understanding.

• P5, L134: Consider removing "For simplicity, $k_y$ and $k_z$ have been set as equal for this model", it has already been mentioned.

• P6, L174: It should be Fig 1.b and not Fig 1.a.

• P7, Fig 1: You could give the title of the colorbox. Is it undisturbed wind speed or wind speed with potential wake effects ?

• P7, L194: Consider writing cos and sin not in italics as for arctan2.

• P13, Eq 12: $\mathrm{d}x$ should be $\mathrm{d}x_i$.

• P17, L323: Consider removing one "the".

• Tables 1/2: Consider rounding the numbers to integral numbers or with one decimal.

• P22, L378: Consider replacing "cause" by "causes".

• P22, L390: Consider replacing "show" by "shows".

**[WESD](WESD)**

---

## Referee Comment (RC2) · Anonymous Referee #2 · 23 Apr 2020

The submitted manuscript outlines a modification to the FLORIS package to allow for heterogeneous "freestream" flow conditions at each turbine in the wind farm, i.e. heterogeneous wind speed, direction, and turbulence intensity at each turbine location if no turbines were present. Improving wake models in heterogeneous flow, where wake model assumptions break down, is critical for the design and controls communities, and therefore the subject matter is of relevance to this journal. While this referee recognizes the challenge of formulating consistent engineering models which satisfy key conservation equations, I have some concerns about the derivation of the heterogeneous wake model which should be revisited and articulated by the authors, as this would establish confidence that the newly proposed method could be applied in a general model setting. Further, the test problem shown lacks enough detail to be

replicated by readers and must be significantly expanded in detail and in explanation, as there are occurrences of model success and failure. Since I believe this model has the potential to be useful to the community, but the manuscript submitted should be modified significantly, I recommend a major revision.

Please also note the supplement to this comment:
https://www.wind-energ-sci-discuss.net/wes-2020-57/wes-2020-57-RC2-supplement.pdf

—————————————————

[Figure]

**Supplement:**

Review of: "Design and analysis of a spatially heterogeneous wake" by Alayna Farrell, Jennifer King, Caroline Draxl, Rafael Mudafort, Nicholas Hamilton, Christopher J. Bay, Paul Fleming, and Eric Simley

**Overall comment:**

The submitted manuscript outlines a modification to the FLORIS package to allow for heterogeneous "freestream" flow conditions at each turbine in the wind farm, i.e. heterogeneous wind speed, direction, and turbulence intensity at each turbine location if no turbines were present. Improving wake models in heterogeneous flow, where wake model assumptions break down, is critical for the design and controls communities, and therefore the subject matter is of relevance to this journal. While this referee recognizes the challenge of formulating consistent engineering models which satisfy key conservation equations, I have some concerns about the derivation of the heterogeneous wake model which should be revisited and articulated by the authors, as this would establish confidence that the newly proposed method could be applied in a general model setting. Further, the test problem shown lacks enough detail to be replicated by readers and must be significantly expanded in detail and in explanation as there are occurrences of model success and failure. Since I believe this model has the potential to be useful to the community, but the manuscript submitted should be modified significantly, I recommend a major revision.

**General comments:**

1. This article would greatly improve with a more formal statement of the research objectives. As discussed in the **Specific comments** points, the Abstract and Introduction are full of comments on issues which affect wake model "accuracy." Wake models are fundamentally low-order and are typically derived from first principles with explicit assumptions (uniform 2D or 3D flow chiefly among them). It would be helpful to consider this more carefully.
   a. Define the objectives of the study and model "accuracy" formally in the introduction. There has to be some degree of baseline performance, since FLORIS cannot be expected to capture power production in strongly complex terrain, for example, since the assumptions made at the stage of derivation break down themselves. Is the hope to capture SCADA power data without resolving any terrain or is the hope to capture realistic flow features (e.g. compare well to LES/WRF in complex terrain)? If the latter is not the goal, how can you demonstrate confidence in the former?
   b. Discuss previous studies which have highlighted issues with uniform inflow formulations and how this study specifically addresses those issues. I recommend expanding the literature review.
   c. Three previous heterogeneous models are discussed, why are those methodologies not employed or inaccurate such that this study is necessary?
2. Problematically, the proposed method does conserve momentum and gives different values of turbine thrust depending on the size of the control volume drawn around the turbine in complex flow (see discussion below). Many engineering models do not

conserve key quantities, but it is important to derive consistent models from first principles otherwise we will not know when the core assumptions are valid or invalid in a new wind farm or model situation.

3. The new methods would benefit from a validation case of the methods (e.g. a comparison to complex flow RANS/LES rather than just comparing power predictions for one wind farm). It's hard to extrapolate that marginally improved power production modeling for one wind farm generalizes to claim that the newly developed model is an improvement given all of the uncertainties associated with low-order wake models and empirical considerations detailed in the **Specific comments** below.

4. There are a significant number of questions/issues with the results section of this manuscript. I have detailed them below in the **Specific Comments**. If the authors can address these comments the manuscript would greatly improve. Very little information/data is given about the test case and even in this limited test scenario the model 'improvement' is not convincing since it does not outperform homogeneous FLORIS for all cases and there is no explanation given for the varying degrees of success.

**Specific comments:**

1. Line 5: The abstract should briefly mention the hypothesized causes of heterogenous wind flow. It is not clear to this referee just by reading the abstract the focus of the heterogenous model. Specifically, is this paper addressing heterogeneity due to: 1) site-specific complex terrain, 2) short-time averaging of quasi-homogeneous turbulent flow, 3) wake heterogeneity, 4) etc. This should be stated concisely in the abstract.

2. Line 10: The abstract should explicitly state the key results of the paper. For example, was the new heterogeneous extension to FLORIS successful in the figures of merit of focus for the present study?

3. Introduction:
   a. The introduction is very brief and can be improved as discussed in **General comments**.
   b. This introduction assumes significant familiarity with FLORIS. That would be acceptable in a conference paper but not for a journal article, which should be self-contained. For example, the concept of "steady state" time-averaging in the FLORIS name isn't even introduced.
   c. The introduction should cover wake models more broadly rather than only FLORIS, since this paper is attempting to develop new heterogeneous wake model capabilities for the literature.

4. Equation 1: Define the axial induction factor

5. Line 95: Since this article details modifications to the flow calculation within FLORIS, the authors should *explicitly detail all* of the assumptions within the derivation of the Gaussian wake model to ensure consistency between the analytical wake model formulation and the freestream condition specification in this implementation of FLORIS. For example, the Guassian wake model (Bastankhah & Porte-Agel (2014)) assumes zero pressure gradients which is then violated in the heterogeneous model.

6. Equation 7: The current proposed method of heterogeneous wind speed, should the local shear coefficient also be modified?
7. Equation 8: How are $k_a$ and $k_b$ affected by complex terrain since the empirical fit to idealized LES calculations performed by Niayifar and Porte Agel assume no terrain
8. Equation 8: It is very unlikely that $k_y = k_z$ in complex terrain. Please perform a sensitivity analysis of the results on this assumption.
9. Equation 10: This equation was also empirically tuned for homogeneous flow and simple terrain and a sensitivity analysis on these parameters must be investigated.
10. Section 3.1 would benefit from a pseudo-code/diagram to improve reader understanding
11. Line 185: The explanation of the selection of interpolation algorithms is insufficient.
    a. What is the justification of linear Barycentric interpolation? Likely the validity of linear interpolation depends on the complexity of the underlying terrain and should be discussed in more detail.
    b. *Detailed* comparisons for the extrapolation should be shown in the Appendix and not just mentioned briefly, since often sensors are not widely available at wind farm sites (usually only a few MET towers for many tens of turbines). Therefore, the performance of the extrapolation will likely be critical to model success.
12. Line 200: The 3D velocity field calculation with the power law assumes that the MET towers are in the same vertical location (z) (otherwise the Barycentric interpolation would not be possible I believe). Often MET masts have varying heights, and this may be of interest to consider for the authors.
13. Figures 1 and 2: What is the color axis representing in the sketch?
14. Line 204: What is $u_\infty$ used by the sum-of-squares velocity deficit in the local velocity calculation in heterogeneous flow?
15. Line 204: **Please state the equation for the velocity deficit update explicitly.** For the purpose of this review, I will assume it is as stated below, although if the formulation is different then this discussion may not apply. I assume that this is the formulation also because Figure 6 has a velocity deficit axis which becomes negative.
    $u(x,y,z) = U_{init} * (1 - C[exp(-(y-\delta)^2/2\sigma_y)exp(-(z-z_h)^2/2\sigma_z)$ where C is a function of the upwind turbine's $C_T$ which is a function of the *average velocity of the upwind turbine*
    $$U_{upwind}$$

    The velocity deficit calculation is not consistent with actuator disk theory since $U_{upwind} \neq U_{init}$ . The velocity deficit trailing a wind turbine ( $u(x,y,z)$ in Equation (4)) is a function of $C_T$ and $U_{upwind}$ . The local calculation here specifies that the velocity deficit trailing a turbine is a function of the turbine thrust coefficient (which is based on the average velocity of the upwind turbine) and the *downwind* velocity.
    a. Illustrative example: A turbine generates a velocity deficit $u_1(x,y,z)$ in a uniform flow field. If in complex terrain, there was a local flow acceleration due to a hill downwind of the turbine, that means the velocity deficit will also increase.

b. This formulation means that the turbine thrust is not a fixed quantity but depends on the downwind position (since $U_{init}$ is a function of x) and therefore momentum is not conserved (as shown by a control volume analysis). Heterogeneities in $U_{init}$ arise from pressure gradients which are neglected in the Gaussian wake model and FLORIS.

c. Perhaps the authors have only used $U_{init}$ in the sum-of-squares calculation?

d. The authors should consider Brogna et al. "A new wake model and comparison of eight algorithms for layout optimization of wind farms in complex terrain" (2020) which proposes a modified Gaussian wake model in complex terrain where the spatial $U_\infty$ evolution is considered in the superposition but not in the velocity deficit calculation aside from modifying the turbine specific $C_T$.

16. Figure 3: No details of the domain geometry, turbines, etc are shown for this figure and it will be very hard to reproduce. It is unclear to this referee what this figure adds, since it shows different velocity colors but it is unclear whether these heterogeneous speeds are valid/correct with no underlying baseline solution (e.g. from complex terrain LES or LiDAR).

17. Figure 5: The colormap is confusing or incorrect since the velocity deficit values are not computed

18. Line 215: Are the 3D velocity deficits (due to the changing inflow angle) included in the local wind direction computation of downwind turbines?

19. Line 225: More discussion of the sensitivity to grid spacing is warranted. What were the authors' methodology for changing the grid spacing in the y-direction?

20. Line 230: The current model does not resolve the momentum source/sinks from the complex terrain and therefore does not satisfy momentum conservation even with a fixed spacing in the y-direction.

21. Line 240: What is the limiting case of wind direction changes that this model can accept?

22. Line 255: The discussion of turbulence intensity's influence on the power curve deserves some literature review, as this has been studied previously (e.g. "Accounting for the effect of turbulence on wind turbine power curves" Clifton & Wagner TORQUE 2014).

23. Figure 10: From this figure, the wake losses look very insignificant due to large streamwise spacing. What would be the power production prediction if the wake model was not used and the power of each turbine was computed only using $U_{init}$?

24. Section 4:

a. The terrain map should be shown given that this paper aims to represent heterogeneity associated with complex terrain/wind flow conditions

b. More details on the SCADA data processing should be given in the Appendix, and ideally, the data would be provided to ensure reproducibility of the results. If the SCADA data must be kept confidential, another test case (with data) must be provided in this paper to ensure reproducibility of results.

c. Why was a timestep of 30 minutes chosen for the FLORIS model runs? Have the authors performed a sensitivity analysis on that timescale selection?

d. Figure 10: The axes are not labeled with the physical coordinates, so the advection time scale of the wind farm cannot be estimated and the results will not be reproducible.

e. Figures 11 and 12: What do the authors refer to as "atmospheric variations?" Figures of the wind speeds, directions, turbulence intensity, etc. should be included for the MET towers, at least in an Appendix.

f. Figures 11 and 12: What has the power been normalized by? No details are given on the normalization strategy.

g. Figure 15: For context, please include a vertical line for each of the cases showing the mean percent error over the datasets overlaid on the histograms.

h. The wind farm power production per turbine should be included as well. This paper gives no indication of the wake losses at the site.

i. Table 1
   i. The heterogeneous model outperforms the homogeneous case when the wind speed is larger (>11 m/s) and there are small wake losses.
   ii. The model also outperforms homogeneous within 5-11 m/s where wake losses are present.
   iii. The authors do not give a clear explanation as to why the model performs poorly in low wind speed (when I assume heterogeneity is more significant at the site but I cannot deduce this from data since that data has not been shown). The authors instead show the results in a different metric and claim success. It would be much more valuable for the community to understand and explain why the new heterogeneous model correction sometimes is good and sometimes is bad at this particular site.

**Technical corrections:**

1. Title: It would be more precise for the title of this manuscript to be "Design and analysis of a wake model for spatially heterogeneous flow**"**

2. Line 90: Period typo

---

## Author Comment (AC1) · 6 Nov 2020

The submitted manuscript addresses the reviewer's comments regarding the article "Design and analysis of a spatially heterogeneous wake", and proposes changes to the original manuscript for improvement. Major revisions have been made to the manuscript, including addressing concerns about the derivation and physical inaccuracies of the heterogeneous wake model, adding more details regarding the operation of the heterogeneous model, providing more thorough analysis of model performance, conducting numerous new simulations to measure performance criteria, and much more. After these major revisions, the manuscript has improved immensely in quality and is ready for re-submission.

[Figure]

The responses to referee comments and the tracked changes to the original article manuscript are attached as supplemental material to this comment.

Please also note the supplement to this comment:
https://wes.copernicus.org/preprints/wes-2020-57/wes-2020-57-AC1-supplement.pdf

---

## Author Response (AR1)

**Response to Referee Comments for**
**"Design and analysis of a spatially heterogeneous wake"**

Corresponding author: Alayna Farrell

November 5, 2020

**Abstract**

The authors would like to thank the referees for their comments. The authors believe that this paper is much improved by addressing the referees' concerns.

**1 Referee 1:**

**General comments:**

This paper presents an interesting improvement of the FLORIS wind farm model with the implementation of a method to take into account an heterogeneous atmospheric inflow. The original wind farm model is well described and efforts have been made on the description of the new implementation with plots that are quite useful for the comprehension, but there are still grey areas and it lacks information about the processing of the z-dimension for the complex terrain application: this application is mentioned twice in the introduction and in the conclusion, and the test case is a wind farm in complex terrain, but no information is given on this specific point. While the test case lacks some detailed information about the wind farm and the atmospheric conditions, a comprehensive comparison has been performed between the original homogeneous FLORIS and the two presented improvements. An exhaustive presentation of quantitative indicators is given and the authors provide well-detailed explanations and conclusions.

Thus, more explanations should be given on the wind direction change processing and the processing of the vertical dimension should also be addressed if a potential application remains "wind farms in complex terrains". Therefore I suggest a major revision.

Here are some general suggestions:

- In the introduction, the authors could mention other state-of-the-art wake modeling utilities and give more general information on the heterogeneity part with reference to studies on the characterization of heterogeneous conditions, the impact of spatial heterogeneity on power predictions...

  *Additional discussion of other wake modeling utilities and the impact of spatial heterogeneity on power prediction accuracy has been added to the introduction. The reader is provided references that offer additional findings related to these topics from Yang et al. (2019) and Clifton and Lundquist (2012) as well.*

- About the description of the new implementation, the authors do not mention how they deal with the vertical dimension, especially since they mention in the introduction that a potential application of this new version is wind farms in complex terrains, and the test case is in complex terrains.

  *The objective of the proposed methods in this study is to capture heterogeneous atmospheric effects caused by site-specific terrain features, without explicitly modeling the geometry of the wind farm terrain. Therefore, the vertical (z) dimension is not considered when interpolating and extrapolating from the atmospheric inputs. Instead, all input values are assumed to be at the same z location, and the interpolation is performed on a two-dimensional plane at this height. Although this approximation may result in a less accurate result, this approach allows the interpolation and extrapolation algorithm to operate with less computational cost. Additional discussion regarding this issue has been added to the abstract and Section 3.1.*

- The part addressing the wind direction heterogeneity and the mesh deformation was not very crystal clear for me and needs more details. Maybe a second case without a constant change in wind direction could be interesting.

  *More details addressing the implementation of heterogeneous wind direction in the model has been added to Section 3.3. A second example of non-constant heterogeneous wind direction simulation in an irregularly spaced wind farm has also been included in this section.*

- About the test case, the authors could give more information about the wind farm(i.e. number and type of turbines, layout/inter-distance), some information about the complexity of the terrain and about the atmospheric measurements at met masts (temporal evolution of wind speed, wind direction and TI for Days A and B, and some information about the stability).

*More information has been added regarding wind farm characteristics. See Section 4, and Appendix A. The chosen wind farm contains several hundred utility-scale wind turbines in location that often is influenced by orographic atmospheric effects. The average stream-wise and span-wise inter-distances are 20D and 2D, respectively, where D represents the average rotor diameter of the turbines in this farm.*

**Specific comments**

- P1, L19: you could mention a reference on the "accurate results in uniform set of atmospheric conditions".

  *A reference was added to a recent study featuring floris: Fleming et al. (2019)*

- P3, L68: you mention "yaw-misalignment conditions" but the $\cos(\gamma)$ is missing in Equation 3. However, as you don't consider any yawing strategy in the paper, maybe you could drop the $\cos(\gamma)$ mentions in all equations. Moreover, $u$ should be infinite velocity or without induction zone.

  *$\cos(\gamma)$ has been added to Eq. 3.*

- In Section 2.3 : you could mention the limitations of the Gaussian model (only valid in far wake)

  - P4, L106: you mention a dependence on ambient TI, but it does not appear until Eq 8. You could mention that this dependence is hidden in k with a reference to Eq 8.

    *This has been noted in the text.*

  - P4, L114: why do you use quadratic superposition of velocity deficits ? You have an added-TI model and you mention Niayifar and Porté-Agel later: in their paper, they recommend the use of linear superposition of velocity deficit while having an added-TI model.

    *Although linear superposition is available as a wake combination method in FLORIS, as discussed in Hamilton et al. (2020), the sum-of-squares method was used in this study because it is a current standard in wake modeling.*

– P5, L140: You could nuance this paragraph on turbulence and saturation effect as it is not well understood for now.

*This was noted in the text.*

– P6, L145: why is the added-TI model part located in the atmospheric stability section ? Moreover, the equation describing the Crespo model is not correct, it should be $0.73a^{0.8325}I_0^{0.0325}(\frac{x}{D})^{-0.32}$. You should also mention the validity ranges ($5 < x/D < 15$, $0.07 < I_0 < 0.014$ and $0.1 < a < 0.4$).

*Although the formula you have listed is the correct classic Crespo model, this equation has been 'tuned' from comparisons to higher fidelity models and field study results in Fleming et al. (2020b) and King et al. (2020) to more accurately capture impacts such as secondary steering, deep-wake effects and yaw-induce wake recovery.*

- In Section 3:

– P6, L152: You could specify that the heterogeneous flows are undisturbed atmospheric flows (i.e. without wake effects).

*This specification has been further emphasized in this section.*

– P6, L158: You could make a reference to Fig 1.a.

*A reference to Fig. 1a has been added.*

– P8, L185: You could name the mentioned algorithms that have been tested for extrapolation, or not mention at all their disadvantages as the explanations are a bit vague and it is difficult to understand what this is about.

*More information regarding specific examples have been added. For example, it was found that the analytic continuation of Radial Basis Functions (RBF) and fitted polynomial splines outside of the initial domain often produced a non-feasible output that did not respect the physical limitations of the atmospheric characteristic being extrapolated.*

– P8, L203: You could mention a reference to Section 3.3. for the processing of wind direction heterogeneity.

*A reference to Section 3.3 has been added to the text.*

– P10, Fig 4 and others: You could name turbines (T1 ... T6) and make a reference to T6 in the text.

*The turbines have been numbered in Fig. 5a and these numbers have been referenced throughout the text.*

– In general for Section 3.3: this procedure with rotation only works if you have uniform lateral change in wind direction ? Maybe you could choose a more complex case for the wind direction change with a bell behaviour or a S-shape. Moreover, how do you define the centre of rotation ? And how do you deal with wake superposition ?

*This procedure works with much more complex cases, but a simple case was provided to show the concept in a format that is easy to understand. Figure 9 showing more complex cases has been added. The centre of rotation is defined as the center of the flow field grid, as depicted in Figure 5b. After the velocity deficit behind each wake is calculated, it is subtracted from the free stream velocity of the flow field using the sum-of-squares method described in Katic et al. (1986).*

– P10-11, Fig 5 and 6: You could distinguish rotated grid points for single turbine and rotated grid points for all turbines (you have deformation for this grid).

*Fig. 6 (formerly Fig. 5) shows the rotated grid points before taking into account the gradual change in wind direction within the simulated flow field. Fig. 7 shows the rotated grid points after the relative changes in wind direction are used to adjust the rotated grid. The rotated gridpoints shown in Fig. 7 are the locations used to calculate the velocity deficit behind turbine T6 in FLORIS. Both Fig. 6 and 7 show stages of the calculation of turbine T6 only. Each turbine wake is calculated independently in its own rotated grid. More details have been added to this section to make this distinction more clear to the reader.*

– – In Section 3.4: in this subsection, I can not really say if you deal with heterogeneous ambient/undisturbed TI. It needs some clarification: do you deal with heterogeneous TI the same way you deal with heterogeneous wind speed ? You could add a plot with the corresponding TI in Fig 8.

*The implementation of heterogeneous TI and heterogeneous wind speed are similar, in that the initial heterogeneous conditions are established throughout the flow field by interpolating from the input values, and*

*then waked conditions are updated continuously throughout FLORIS computations of flow-field interactions. Calculations for wake propagation use the value of waked turbulence intensity at each turbine, based on the added turbulence model discussed in Section 2.3.2 and Niayifar and Porté-Agel (2015).*

– In Section 3.5: Have you used an aero-elastic solver for this part ? You could also give an order of magnitude for $\Lambda$.

*An aero-elastic solver was not used. $\Lambda$ cannot have a value equal to or less than zero. This has been noted in Subsection 3.5.*

– P14, L276: You could nuance this comment because having an improved Ct should be as important as having an improved Cp as the velocity deficit model relies on Ct.

*This comment has been revised in the paper to highlight the benefit of a possible $C_T$ turbulence-correction method implemented for the calculation of velocity deficit.*

• In Section 4:

– P14, L281: More information could be given on the wind farm, the layout (min/max inter-distance), the turbines, the complexity of the terrain...

*Information regarding the characteristics of the physical layout of the wind farm have been added to Appendix A and Section 4.*

– P16, Figure 11/12: You could give more information on the daily evolution of wind speed, wind direction and TI, and on $\alpha_s$ to have an information about stability.

*This information cannot be provided because it is confidential. It should be noted that the analysis of Day B was removed because it did not add much to the discussion. This study was primarily focused on improving the power output forecast of conditions that are more variant, and day B was comparatively less variant than day A.*

• In Section 5:

– P22, L389: You could give an approximate value of the power prediction improvement.

*The percent reduction in mean absolute error for the overall wind farm (14.6% for the heterogeneous model, and 31.42% for the heterogeneous model with turbulence correction) has been added to the conclusion. It should be noted that this improvement factor will vary depending on many circumstances, including the weather patterns and physical layout characteristics a specific wind farm site.*

**Technical corrections**

- In general with the plots on wind direction changes, you could add one or two streamlines, it could help in the understanding.

  *White line contours are included on the plots, which help in outlining the wake to increase visibility of flow patterns.*

- P5, L134: Consider removing "For simplicity, ky and kz have been set as equal for this model", it has already been mentioned.

  *This statement has been removed.*

- P6, L174: It should be Fig 1.b and not Fig 1.a.

  *This typo has been fixed.*

- P7, Fig 1: You could give the title of the colorbox. Is it undisturbed wind speed or wind speed with potential wake effects ?

  *To eliminate confusion, Figures 2 and 3 now show an interpolation performed for initial undisturbed wind speed specifically. A label for the colorbar has been added to the figures for reference.*

- P7, L194: Consider writing cos and sin not in italics as for arctan2.

  *This change has been made.*

- P13, Eq 12: $d_x$ should be $d_{xi}$.

  *This change has been made.*

- P17, L323: Consider removing one "the".

  *This change has been made.*

- Tables 1/2: Consider rounding the numbers to integral numbers or with one decimal.

  *This change has been made.*

- P22, L378: Consider replacing "cause" by "causes".

  *This change has been made.*

- P22, L390: Consider replacing "show" by "shows".

  *This change has been made.*

**2 Referee 2:**

**Overall Comment:**

The submitted manuscript outlines a modification to the FLORIS package to allow for heterogeneous "freestream" flow conditions at each turbine in the wind farm, i.e. heterogeneous wind speed, direction, and turbulence intensity at each turbine location if no turbines were present. Improving wake models in heterogeneous flow, where wake model assumptions break down, is critical for the design and controls communities, and therefore the subject matter is of relevance to this journal. While this referee recognizes the challenge of formulating consistent engineering models which satisfy key conservation equations, I have some concerns about the derivation of the heterogeneous wake model which should be revisited and articulated by the authors, as this would establish confidence that the newly proposed method could be applied in a general model setting. Further, the test problem shown lacks enough detail to be replicated by readers and must be significantly expanded in detail and in explanation as there are occurrences of model success and failure. Since I believe this model has the potential to be useful to the community, but the manuscript submitted should be modified significantly, I recommend a major revision.

**General Comments:**

1. This article would greatly improve with a more formal statement of the research objectives. As discussed in the **Specific comments** points, the Abstract and Introduction are full of comments on issues which affect wake model "accuracy." Wake models are fundamentally low-order and are typically derived from first principles with explicit assumptions (uniform 2D or 3D flow chiefly among them). It would be helpful to consider this more carefully.

    (a) Define the objectives of the study and model "accuracy" formally in the introduction. There has to be some degree of baseline performance, since FLORIS cannot be expected to capture power production in strongly complex terrain, for example, since the assumptions made at the stage of derivation break down themselves. Is the hope to capture SCADA power data without resolving any terrain or is the hope to capture realistic flow features (e.g. compare well to LES/WRF in complex terrain)? If the latter is not the goal, how can you demonstrate confidence in the former?

    *In developing this proposed model, the objective was to capture a more accurate representation of the effects of wind farm wake interactions within complex terrain, without actually resolving any terrain geometry during simulation. This study analyzes the heterogeneous model's accuracy in power output prediction as a measure of FLORIS mod-*

*eling performance. Additional discussion regarding the objectives of this study have been added to the introduction.*

(b) Discuss previous studies which have highlighted issues with uniform inflow formulations and how this study specifically addresses those issues. I recommend expanding the literature review.

*Further discussion of literature that investigates this issue (such as Yang et al. (2019)) has been added to the introduction.*

(c) Three previous heterogeneous models are discussed, why are those methodologies not employed or inaccurate such that this study is necessary?

*The other heterogeneous models were mentioned in this paper to acknowledge their efforts in the body of research related to this issue of spatial heterogeneity within wind farms in complex terrain. Each of these referenced models present valid and useful findings, but their methods were not employed in this study because a potential for success was also identified using the approach of the proposed model. This study was conducted to specifically analyze the effectiveness of the proposed approach to modeling spatially heterogeneous flow.*

2. Problematically, the proposed method does conserve momentum and gives different values of turbine thrust depending on the size of the control volume drawn around the turbine in complex flow (see discussion below). Many engineering models do not conserve key quantities, but it is important to derive consistent models from first principles otherwise we will not know when the core assumptions are valid or invalid in a new wind farm or model situation.

*These issues have been addressed in the paper and in the relevant* **Specific comments** *below.*

3. The new methods would benefit from a validation case of the methods (e.g. a comparison to complex flow RANS/LES rather than just comparing power predictions for one wind farm). It's hard to extrapolate that marginally improved power production modeling for one wind farm generalizes to claim that the newly developed model is an improvement given all of the uncertainties associated with low-order wake models and empirical considerations detailed in the **Specific comments** below.

*In another recent study (Fleming et al., 2020a), simulations from the*

*proposed heterogeneous FLORIS model are compared to LES results for a case study of a 38-turbine wind farm. The analysis showed that including spatially heterogeneous wind speed lowers the error in predictions of total power production by 15% in comparison to the homogeneous solution. Further comments regarding validations in this study have been addressed in relevant **Specific comments** below.*

4. There are a significant number of questions/issues with the results section of this manuscript. I have detailed them below in the **Specific Comments**. If the authors can address these comments the manuscript would greatly improve. Very little information/data is given about the test case and even in this limited test scenario the model 'improvement' is not convincing since it does not outperform homogeneous FLORIS for all cases and there is no explanation given for the varying degrees of success.

   *These issues have been addressed in the relevant **Specific comments** below.*

**2.1   Specific comments:**

1. Line 5: The abstract should briefly mention the hypothesized causes of heterogenous wind flow. It is not clear to this referee just by reading the abstract the focus of the heterogenous model. Specifically, is this paper addressing heterogeneity due to: 1) site-specific complex terrain, 2) short-time averaging of quasi-homogeneous turbulent flow, 3) wake heterogeneity, 4) etc. This should be stated concisely in the abstract.

   *The objective of the proposed methods is to capture heterogeneous atmospheric effects caused by site-specific terrain features, without explicitly modeling the geometry of the wind farm terrain. This has been mentioned in the abstract.*

2. Line 10: The abstract should explicitly state the key results of the paper. For example, was the new heterogeneous extension to FLORIS successful in the figures of merit of focus for the present study?

   *Information regarding this model's performance has been added to the abstract.*

3. Introduction:

   (a) The introduction is very brief and can be improved as discussed in **General comments.**

*See responses to No. 1 in **General Comments.***

(b) This introduction assumes significant familiarity with FLORIS. That would be acceptable in a conference paper but not for a journal article, which should be self-contained. For example, the concept of "steady state" time-averaging in the FLORIS name isn't even introduced.

*Additional information regarding the concept of "steady state" time-averaging and other general details about FLORIS have been added to Section 2. The reader is also given several sources to expand further background research on the basic concepts involving FLORIS operation.*

(c) The introduction should cover wake models more broadly rather than only FLORIS, since this paper is attempting to develop new heterogeneous wake model capabilities for the literature.

*Additional references to other wake models has been added to the introduction.*

4. Equation 1: Define the axial induction factor

*A definition of the axial induction factor has been added.*

5. Line 95: Since this article details modifications to the flow calculation within FLORIS, the authors should explicitly detail all of the assumptions within the derivation of the Gaussian wake model to ensure consistency between the analytical wake model formulation and the freestream condition specification in this implementation of FLORIS. For example, the Guassian wake model (Bastankhah & Porte-Agel (2014)) assumes zero pressure gradients which is then violated in the heterogeneous model.

*Additional discussion relating to the assumptions within the derivation of the Gaussian wake model, and the ways in which the heterogeneous methods may violate these assumptions have added in Section 2.3.1*

6. Equation 7: The current proposed method of heterogeneous wind speed, should the local shear coefficient also be modified?

*In this study, the shear coefficient was assumed to be homogeneous throughout the flow field. FLORIS currently does not have functionality to define*

*a spatially heterogeneous shear coefficient. In future work, the benefit of this functionality may be investigated for improvement of the model.*

7. Equation 8: How are ka and kb affected by complex terrain since the empirical fit to idealized LES calculations performed by Niayifar and Porte Agel assume no terrain

    *In Fleming et al. (2020b), the results of a field study analysis focusing on the performance of these tuned parameters is presented, comparing two campaigns located in comparatively simple and complex terrains. The results of this study show a trend of possible underprediction of wake losses in areas with complex terrain due to the influence of several tuning parameters. Since the same default values for $k_a$ and $k_b$ are used in Fleming et al. (2020b) as in this study, it provides a very relevant analysis of the fit of these terms.*

8. Equation 8: It is very unlikely that ky= kz in complex terrain. Please perform a sensitivity analysis of the results on this assumption.

    *Based on the defined relationship between $k_y/k_z$ and $k_a/k_b$, it would be reasonable to assume that the effects that complex terrain have on $k_y$ and $k_z$ will be similar to those observed for $k_a$ and $k_b$ in Fleming et al. (2020b).*

9. Equation 10: This equation was also empirically tuned for homogeneous flow and simple terrain and a sensitivity analysis on these parameters must be investigated.

    *The findings from Fleming et al. (2020b) indicate that there may be a slightly worsened effect of FLORIS's tendency to underpredict wake losses in areas with complex terrain using this equation. These effects have been noted in this section of the paper, and the parameters in this equation have also been updated to reflect the most recent tunings found in FLORIS currently.*

10. Section 3.1 would benefit from a pseudo-code/diagram to improve reader understanding

    *A psuedo-code diagram has been added to explain the steps involved in initializing the flow field. See Fig. 1.*

11. Line 185: The explanation of the selection of interpolation algorithms is insufficient.

(a) What is the justification of linear Barycentric interpolation? Likely the validity of linear interpolation depends on the complexity of the underlying terrain and should be discussed in more detail.

*Linear Barycentric interpolation was chosen because it is relatively simple in computation and can be easily implemented without requiring any input parameters other than the locations and values of wind measurements. It is correct that the accuracy of the interpolated values is dependent on the input measurements provided, and the complexity of the weather patterns in the physical wind farm. Additional information regarding this issue has been added to Section 3.1.*

(b) Detailed comparisons for the extrapolation should be shown in the Appendix and not just mentioned briefly, since often sensors are not widely available at wind farm sites (usually only a few MET towers for many tens of turbines). Therefore, the performance of the extrapolation will likely be critical to model success.

*For this test case, it is difficult to compare the accuracy of extrapolated wind measurements, because there is little data available that shows the actual atmospheric behavior at the wind farm during the time span of this study, other than the MET masts used for inputs for the FLORIS simulations. The heterogeneous model is not able to introduce precise details of initial flow-field conditions beyond the bounds of the measurements taken from the wind farm, but this is not the goal of the extrapolation processes. These extrapolations within the heterogeneous model aim to represent an approximation of the observed conditions, based on the limited wind measurements provided.*

12. Line 200: The 3D velocity field calculation with the power law assumes that the MET towers are in the same vertical location (z) (otherwise the Barycentric interpolation would not be possible I believe). Often MET masts have varying heights, and this may be of interest to consider for the authors.

*In this study, all input measurement locations were assumed to be at the same vertical location (z) for simplicity. In future work, it may be beneficial to add the functionality of varying input measurements to FLORIS, given that MET towers are typically located at varying heights in real wind farms.*

13. Figures 1 and 2: What is the color axis representing in the sketch?

*For clarity, Fig. 1 and 2 (now Fig. 2 and 3) now indicate the interpolation*

*performed for wind speed, in meters per second. A label for the colorbar has been added to the figures for reference.*

14. Line 204: What is u used by the sum-of-squares velocity deficit in the local velocity $u_\infty$ calculation in heterogeneous flow?

    *The value of $U_\infty$ used in the deficit calculation for $u(x, y, z)$ in Eq. 4 is equal to $U_{init}$, which is calculated using the power-log law of wind (Eq. 7).*

15. Line 204: **Please state the equation for the velocity deficit update explicitly.** For the purpose of this review, I will assume it is as stated below, although if the formulation is different then this discussion may not apply. I assume that this is the formulation also because Figure 6 has a velocity deficit axis which becomes negative. $u(x, y, z) = U_{init} * (1 - C[exp(-(y - \delta)^2/2\sigma_y)exp(z - z_h)^2/2\sigma_z)$ where C is a function of the upwind turbine's $C_T$ which is a function of the average velocity of the upwind turbine $U_{upwind}$. The velocity deficit calculation is not consistent with actuator disk theory since $U_{upwind} \neq U_{init}$. The velocity deficit trailing a wind turbine ( $u(x, y, z)$ in Equation (4)) is a function of $C_T$ and $U_{upwind}$. The local calculation here specifies that the velocity deficit trailing a turbine is a function of the turbine thrust coefficient (which is based on the average velocity of the upwind turbine) and the *downwind velocity*.

    *The equation for velocity deficit that is stated above is the one used in the proposed model. Section 2.2 has been revised so that this equation is explicitly stated.*

    (a) Illustrative example: A turbine generates a velocity deficit $u_1(x, y, z)$ in a uniform flow field. If in complex terrain, there was a local flow acceleration due to a hill downwind of the turbine, that means the velocity deficit will also increase.

    (b) This formulation means that the turbine thrust is not a fixed quantity but depends on the downwind position (since $U_{init}$ is a function of x) and therefore momentum is not conserved (as shown by a control volume analysis). Heterogeneities in $U_{init}$ arise from pressure gradients which are neglected in the Gaussian wake model and FLORIS.

    (c) Perhaps the authors have only used $U_{init}$ in the sum-of-squares calculation?

    *$U_{init}$ has been used in the velocity deficit calculations, and in the sum-of-squares calculations. This means that the proposed heterogeneous violates the principle of momentum conservation, according to this control-volume analysis. This has been noted in the paper.*

(d) The authors should consider Brogna et al. "A new wake model and comparison of eight algorithms for layout optimization of wind farms in complex terrain" (2020) which proposes a modified Gaussian wake model in complex terrain where the spatial $U_\infty$ evolution is considered in the superposition but not in the velocity deficit calculation aside from modifying the turbine specific $C_T$.

*In future developments, the benefits of an approach similar to this may be investigated to improve overall momentum conservation in the FLORIS model. This has been mentioned in the paper and the relevant article cited.*

16. Figure 3: No details of the domain geometry, turbines, etc are shown for this figure and it will be very hard to reproduce. It is unclear to this referee what this figure adds, since it shows different velocity colors but it is unclear whether these heterogeneous speeds are valid/correct with no underlying baseline solution (e.g. from complex terrain LES or LiDAR).

*Fig. 3 (now Fig. 4) provides an exemplary hypothetical case to show how the heterogeneous effects of the model can be observed visually as an additional method of analysis. These plots have not been compared to LES or LiDAR solutions.*

17. Figure 5: The colormap is confusing or incorrect since the velocity deficit values are not computed

*The velocity deficit colorbar has been removed from this figure for clarity.*

18. Line 215: Are the 3D velocity deficits (due to the changing inflow angle) included in the local wind direction computation of downwind turbines?

*The velocity deficit of upwind turbines does not affect the local wind direction at a downwind turbine. The wind direction at all points in the flow field are interpolated from the initial inputs when the flow field is initialized. After flow field initialization, the wind direction values are not changed during wake calculations, unless the flow field is re-initialized with differing wind direction inputs.*

19. Line 225: More discussion of the sensitivity to grid spacing is warranted. What were the authors' methodology for changing the grid spacing in the y-direction?

*As discussed in Section 3.4, the changes in the grid spacing in the y-direction are dependent on the varying gradient of wind direction within the flow field. Without this variation, the FLORIS model would only be able to define a single wind direction for each turbine, and not a gradual change throughout the flow domain.*

20. Line 230: The current model does not resolve the momentum source/sinks from the complex terrain and therefore does not satisfy momentum conservation even with a fixed spacing in the y-direction.

    *This is an important concept to consider in this proposed model, and has been noted in the paper. In the heterogeneous model, the wind farm's physical terrain is not modeled, and the effects of this terrain are approximated to the heterogeneous wind measurements used as inputs. Although this method does not conserve momentum, the approximations imposed in the model prove to be effective in modeling the effects of the landscape, based on the results of this study.*

21. Line 240: What is the limiting case of wind direction changes that this model can accept?

    *The limiting case of wind direction change is that which causes the flow-field grid points to overlap themselves in the process of rotation during velocity deficit calculations, as shown in Fig. 6 and 7. This limit is determined for each farm independently and varies with the site-specific layout geometry of each case. This has also been further elaborated on in Section 3.3.*

22. Line 255: The discussion of turbulence intensity's influence on the power curve deserves some literature review, as this has been studied previously (e.g. "Accounting for the effect of turbulence on wind turbine power curves" Clifton & Wagner TORQUE 2014).

    *The proposed method of accounting for turbulence intensity power effects was developed with the goal of operating without a dependency on the availability of training data or empirical values for this study. In future work, it may be advantageous to incorporate more complex techniques that are able to capture the effects of turbulence intensity with greater detail and accuracy. More information regarding this issue has been added to Section 3.5 of the paper.*

23. Figure 10: From this figure, the wake losses look very insignificant due to large streamwise spacing. What would be the power production prediction if the wake model was not used and the power of each turbine was computed only using $U_{init}$ ?

*Information regarding the relative size and geometry of the wind farm layout and turbines has been added to Appendix A. This should give an indication of the influence of wake effects at this site. Identical simulations were also performed with the omission of wake calculations to evaluate the significance of wake losses for this study. See relevant tables in Appendix B.*

24. Section 4:

   (a) The terrain map should be shown given that this paper aims to represent heterogeneity associated with complex terrain/wind flow conditions

   *The exact terrain map for this study cannot be given because it is proprietary, although the relevant characteristics of the terrain geometry have been added to the Appendix.*

   (b) More details on the SCADA data processing should be given in the Appendix,and ideally, the data would be provided to ensure reproducibility of the results. If the SCADA data must be kept confidential, another test case (with data) must be provided in this paper to ensure reproducibility of results.

   *Unfortunately, the SCADA data is confidential for this wind farm. A second wind farm with similar terrain characteristics and operational conditions could not be found to perform a second simulation, due to the common industry practice of making commercial wind farm SCADA data confidential.*

   (c) Why was a timestep of 30 minutes chosen for the FLORIS model runs? Have the authors performed a sensitivity analysis on that timescale selection?

   *In preliminary time scale sensitivity analyses, it was observed that time steps that were of a longer duration typically had a lower accuracy in power predictions throughout all three FLORIS models.30 minute time steps were originally chosen for this study because they are frequent enough to show substantial exemplification of the added power prediction accuracy offered by the proposed model, while still maintaining a moderate computational expense in simulation. In the recently updated results of this study, the simulations were performed*

*at 10 minute time steps to provide an even more detailed analysis of this model, and to meet the common standards of the industry.*

(d) Figure 10: The axes are not labeled with the physical coordinates, so the advection time scale of the wind farm cannot be estimated and the results will not be reproducible.

*The exact coordinates of the wind farm cannot be given, because this information is confidential, but details that give the relative geometry of the wind farm have been added to the Appendix.*

(e) Figures 11 and 12: What do the authors refer to as "atmospheric variations?" Figures of the wind speeds, directions, turbulence intensity, etc. should be included for the MET towers, at least in an Appendix.

*The "atmospheric variations" discussed in these figures refer to changes in wind direction, wind speed, and turbulence intensity in each given day. This information cannot be included because it is confidential. It should also be noted that the analysis of Day B was removed because it did not add much to the discussion of model performance. This study was primarily focused on improving the power output forecast of conditions that are more variant, and day B was comparatively less variant than day A.*

(f) Figures 11 and 12: What has the power been normalized by? No details are given on the normalization strategy.

*The power has been normalized by the rated output for the subject wind farm. This is now noted in the paper.*

(g) Figure 15: For context, please include a vertical line for each of the cases showing the mean percent error over the datasets overlaid on the histograms.

*Vertical lines have been added to the figures.*

(h) The wind farm power production per turbine should be included as well. This paper gives no indication of the wake losses at the site.

*Table 3 has been added to show the average error from all of the individual turbines of the wind farm. Additional information regarding the relative geometry of the subject wind farm has also been appended to give a better indication of the wake losses at the site.*

(i) Table 1

    i. The heterogeneous model outperforms the homogeneous case when the wind speed is larger (>11 m/s) and there are small wake losses.

    ii. The model also outperforms homogeneous within 5-11 m/s where wake losses are present.

    iii. The authors do not give a clear explanation as to why the model performs poorly in low wind speed (when I assume heterogeneity is more significant at the site but I cannot deduce this from data since that data has not been shown). The authors instead show the results in a different metric and claim success. It would be much more valuable for the community to understand and explain why the new heterogeneous model correction sometimes is good and sometimes is bad at this particular site.

*As discussed in Section 4, this may be due to the inherent "bias" of the metric of Mean Average Percent Error (MAPE), which penalizes overpredictions with more weight than underpredictions. In comparison to the metric Mean Absolute Error (MAE), MAPE also shows an equally weighted mean error regardless of the overall power output per time step, which is often times not preferred for an indication of overall farm power output accuracy. It is possible that the reported increase in MAPE with lower wind speeds may be an indication that the heterogeneous and turbulence intensity correction models tend to produce more frequent overpredictions of power output in conditions where wind speeds are near the cut-in speed.*

*If this is true, it may indicate that the proposed interpolation methods have a tendency to define disproportionately high velocity values ($U_{init}$) at flow-field points in predominantly low-velocity settings, causing an overestimate of power production as a result. In future work, this could be circumvented by implementing more complex interpolation strategies that consider the physical dynamics of changing velocity in naturally occurring fluid flow.*

**Technical corrections:**

1. Title: It would be more precise for the title of this manuscript to be "Design and analysis of a wake model for spatially heterogeneous flow"

*The title has been changed.*

2. Line 90: Period typo.

*This typo has been fixed.*

**References**

[revised manuscript text omitted]

---

## Referee Report (RR1)

General comments:

This paper presents an interesting improvement of the FLORIS wind farm model with the implementation of a method to take into account an heterogeneous atmospheric inflow. The original wind farm model is well described and a considerable effort has been made on the description of the new implementation during the reviewing process. In general, the comments of the last reviewing process were well addressed: the procedure is now more detailed with figures helping to the understanding. The authors propose some elements to discuss the limitations of the model. The test case is well described and the analyse is exhaustive with interesting comments for each metric.

Here are some specific comments and technical corrections:

- In general in the introduction, the authors should be more specific while mentioning "variant conditions". "Spatially variant conditions" is more adapted in order to avoid confusion with unsteady conditions.
    o L37: consider replacing "during these conditions" with "under these conditions".
    o L48: consider adding "spatially" variant weather conditions.
- In Section 3.1
    o Consider adding x/y coordinate axes (or Easting/Northing) in Figures 2 and 3.
    o L236: consider adding a reference to L245 since, at this point of the article, the reader could wonder about how the model deals with different hub heights, especially for wind farms in complex terrain.
- In Section 3.3
    o L256: maybe changing "center of the flow field" into "center of the simulation domain" would make the location of the rotation center more clear ?
    o L286: The sentence is not clear "is that which causes".
- In Section 4
    o There is only one subsection (4.1).
    o L397: Consider adding "with respect to the"
    o L398: Consider changing the end of the sentence "the addition[…] contributes to improvements or improving ?"
    o L441: "is observed"
- In Section 5
    o L476: "indicate"

---

## Referee Report (RR2)

Review of: "Design and analysis of a wake model for spatially heterogeneous flow" by Alayna Farrell, Jennifer King, Caroline Draxl, Rafael Mudafort, Nicholas Hamilton, Christopher J. Bay, Paul Fleming, and Eric Simley

**Overall comment:**

Thanks to the authors for thoroughly addressing my points and performing a major revision. I appreciate the time the authors spent to consider each point in both of the reviews. The discussion of the model, its motivation, and derivation is much clearer to this referee with the revisions made. However, I still have uncertainty about the wind farm data comparison and I recommend another revision to address these questions.

**General comments:**

1. Overall, given the lack of detail (due to understandable IP constraints), the field SCADA data comparison is not ideal for proving that the new model has addressed the issue of flow field heterogeneity. In fact, the most convincing use case of the heterogeneous wake model proposed is presented in a separate paper [1]. This referee recommends including the LES test case in this manuscript.

2. Related to Point 1, there still appear to be unexpected results in the comparison with SCADA data that require more investigation and/or explanation. The proposed method does seem to have good merit, and in some cases comes with substantial improvement compared to the homogeneous model (by getting localized $u_\infty$ estimate improvements), but less attention is given to situations where performance is not affected or worse than homogeneous FLORIS. It would be hard for a FLORIS or other wake model user in the community to understand when to use the heterogeneous model versus the homogeneous in a general model setting based on this paper, especially given the poor performance at low wind speeds.

**Specific comments:**

1. Abstract: The addition of quantitative results in the abstract is helpful, but the discussion has been made selectively. The heterogeneous model does reduce MAE but in fact it increases MAPE and this should be mentioned in the abstract to not appear to be selective by the authors. Based on the discussion, whether the method improves MAE or MAPE compared to homogeneous methods will likely be site-specific (e.g. based on the wind rose) due to poor performance for low wind speeds.

2. Line 28: The reference to Schreiber seems out of place. Data-driven wake model parameter corrections have also been proposed by [e.g. 2,3,4], among others.

3. Brogna *et al*. (2020) should be discussed in the introduction as prior work in the domain.

4. Equation 3: Missing parenthesis.

5. Line 294: What does it mean for "the flow-field grid points to conflict in the rotated grid?" I don't quite understand this sentence or the stated rotation limiting case. Would this be a case where the rotated grid folds back on itself and has different original points in the same rotated x-y space?

6. The added discussion of the TI model is helpful!

7. Figure 13: Can the FLORIS predictions without wake losses be added to this figure for visual comparison?

8. Some of the newly added sentences have typographical errors (e.g. Line 295), I suggest the authors check over them in detail.

9. Thank you to the authors for including the results of FLORIS without wake losses included. I want to be sure I understand the results you're presenting.

    a. Comparing Tables 3 and B2:

        i. The wake losses make no difference to turbine specific MAE for velocity of <5 m/s (expected)

        ii. Including wake losses reduces FLORIS MAE only slightly for 5-11 m/s for the heterogeneous model and has no impact on the homogeneous model (unexpected)

        iii. Including wake losses significantly reduces MAE for >11 m/s where we would not expect significant wake interactions (as the rated wind speed is reached), this is also unexpected. There is no impact for the homogeneous model and more impact for the heterogeneous model.

    b. Comparing Tables 2 and B1, including wake loss modeling degrades FLORIS's performance in the heterogeneous model.

    c. Comparing Tables B1 and B2, why does the heterogeneous model have lower MAE for the farm than homogeneous but higher MAE than homogeneous when MAE is turbine specific?

    d. Given that including or excluding wake effects in FLORIS seems to have a very small impact on the MAE (there is virtually no impact on homogeneous FLORIS MAE), that seems to indicate to this referee that this is not an ideal test case for a wake model.

**References**

1. Fleming, Paul, et al. "Overview of FLORIS updates." *Journal of Physics: Conference Series*. Vol. 1618. No. 2. IOP Publishing, 2020.
2. Teng, Jian, and Corey D. Markfort. "A Calibration Procedure for an Analytical Wake Model Using Wind Farm Operational Data." *Energies* 13.14 (2020): 3537.
3. Howland, Michael F., et al. "Optimal closed-loop wake steering–Part 1: Conventionally neutral atmospheric boundary layer conditions." *Wind Energy Science* 5.4 (2020): 1315-1338.
4. Shapiro, Carl R., et al. "A Wake Modeling Paradigm for Wind Farm Design and Control." *Energies* 12.15 (2019): 2956.

---

## Referee Report (RR3)

**WES-2020-57**

Thanks to the authors for including the FLORIS comparison with LES. These results appear to be more compelling to indicate improvements with the heterogeneous flow wake modeling scheme. I have a few minor comments focusing on improving the clarity of the presented results.

1. There are some typographical errors in the new sentences (e.g. Line 362). I encourage the authors to carefully proofread the paper before finalizing it for publication.

2. Eq. 3: The exponent 'p' should be on the cosine, not on the \gamma. $cos^p(\gamma)$.

3. Line 389, sentence starting with: "The comparison of this metric…"

I assume "this metric" refers to Eq. 4. Figure 14 appears to plot an absolute error for each wind direction which is not the same as MAE defined in Eq. 4, which is an average over all wind directions and would produce only one value, not a plot with wind direction dependency. I believe later, this is defined as "average mean absolute error." It would improve the clarity of the paper to define the quantities explicitly and to be consistent throughout the results section.

I would also suggest that Figure 14 does not show that the heterogeneous model provides an improvement in predictions, but Figure 15 clearly does.

4. Figure 15 is an enlightening result which clearly shows the benefit of turbine specific MAE instead of total wind farm power comparisons and the utility of the proposed model.

5. I suggest the authors clarify their quantity of interest notations (e.g. MAE) and descriptions in the results section which appear to be reused and conflicting. In Eq. 4, MAE is defined as the absolute error in wind farm power production averaged over all the wind direction cases. However, in Figure 15, MAE is defined as the absolute error in wind turbine power production averaged over all the turbines in the wind farm.

6. Figure 15: Caption states "average absolute error" and y-axis states "mean absolute error."

---

## Author Response (AR2)

**Response to Referee Comments for**
**"Design and analysis of a wake model for spatially heterogeneous flow"**

Corresponding author: Alayna Farrell

February 2, 2021

**Abstract**

The authors would like to thank the referees once again for their comments. The authors believe that this paper is further improved by addressing the referees' concerns.

**Referee Report 1:**

**General comments:**

This paper presents an interesting improvement of the FLORIS wind farm model with the implementation of a method to take into account an heterogeneous atmospheric inflow. The original wind farm model is well described and a considerable effort has been made on the description of the new implementation during the reviewing process. In general, the comments of the last reviewing process were well addressed: the procedure is now more detailed with figures helping to the understanding. The authors propose some elements to discuss the limitations of the model. The test case is well described and the analyse is exhaustive with interesting comments for each metric.

Here are some specific comments and technical corrections:

- In general in the introduction, the authors should be more specific while mentioning "variant conditions". "Spatially variant conditions" is more adapted in order to avoid confusion with unsteady conditions.

    ○ L37: consider replacing "during these conditions" with "under these conditions".

    *This edit has been made.*

○ L48: consider adding "spatially" variant weather conditions.

*This edit has been made.*

- In Section 3.1

○ Consider adding x/y coordinate axes (or Easting/Northing) in Figures 2 and 3.

*Axis labels have been added to Figures 2 and 3.*

○ L236: consider adding a reference to L245 since, at this point of the article, the reader could wonder about how the model deals with different hub heights, especially for wind farms in complex terrain.

*A reference has been added.*

- In Section 3.3

○ L256: maybe changing "center of the flow field" into "center of the simulation domain" would make the location of the rotation center more clear ?

*This change has been made.*

○ L286: The sentence is not clear "is that which causes".

*This paragraph has been reworded to improve clarity.*

- In Section 4

○ There is only one subsection (4.1).

*An additional subsection has been added to this section.*

○ L397: Consider adding "with respect to the"

*This edit has been made.*

○ L398: Consider changing the end of the sentence "the addition[...] contributes to improvements or improving ?"

*This edit has been made.*

○ L441: "is observed"

*This edit has been made.*

- In Section 5

○ L476: "indicate"

*This edit has been made.*

**Referee Report 2:**

**Overall Comment:**

Thanks to the authors for thoroughly addressing my points and performing a major revision. I appreciate the time the authors spent to consider each point in both of the reviews. The discussion of the model, its motivation, and derivation is much clearer to this referee with the revisions made. However, I still have uncertainty about the wind farm data comparison and I recommend another revision to address these questions.

**General comments:**

1. Overall, given the lack of detail (due to understandable IP constraints), the field SCADA data comparison is not ideal for proving that the new model has addressed the issue of flow field heterogeneity. In fact, the most convincing use case of the heterogeneous wake model proposed is presented in a separate paper [1]. This referee recommends including the LES test case in this manuscript.

   *An additional Subsection 4.1 has been added to the validation analysis to discuss the accuracy of FLORIS simulations compared to LES, similar to the work of Fleming et al. (2020).*

2. Related to Point 1, there still appear to be unexpected results in the comparison with SCADA data that require more investigation and/or explanation. The proposed method does seem to have good merit, and in some cases comes with substantial improvement compared to the homogeneous model (by getting localized estimate $u_\infty$ improvements),but less attention is given to situations where performance is not affected or worse than homogeneous FLORIS. It would be hard for a FLORIS or other wake model user in the community to understand when to use the heterogeneous model versus the homogeneous in a general model setting based on this paper, especially given the poor performance at low wind speeds.

   *More discussion of uncertainties regarding the model's performance has been added to the conclusion.*

**Specific Comments:**

1. Abstract: The addition of quantitative results in the abstract is helpful, but the discussion has been made selectively. The heterogeneous model does reduce MAE but in fact it increases MAPE and this should be mentioned in the abstract to not appear to be selective by the authors. Based

on the discussion, whether the method improves MAE or MAPE compared to homogeneous methods will likely be site-specific (e.g. based on the wind rose) due to poor performance for low wind speeds.

*A statement regarding the variability of the proposed model's performance in different site-specific operational conditions is now included in the abstract.*

2. Line 28: The reference to Schreiber seems out of place. Data-driven wake model parameter corrections have also been proposed by [e.g. 2,3,4], among others.

   *Discussion of these modeling techniques has been revised and relocated to the next paragraph.*

3. Brogna et al. (2020) should be discussed in the introduction as prior work in the domain.

   *A reference to Brogna et al. (2020) has been added to the discussion of prior work in the introduction.*

4. Equation 3: Missing parenthesis.

   *Parentheses have been added.*

5. Line 294: What does it mean for "the flow-field grid points to conflict in the rotated grid?" I don't quite understand this sentence or the stated rotation limiting case. Would this be a case where the rotated grid folds back on itself and has different original points in the same rotated x-y space?

   *Your explanation of this concept is correct. This refers to a case where the rotated grid points (shown in Fig. 7a) fold back onto themselves, causing the overlapping points to be erroneously assigned velocity deficit if they overlap a region that is in the wake downstream of a turbine. The referenced paragraph in the text has been reworded for clarity.*

6. The added discussion of the TI model is helpful!

7. Figure 13: Can the FLORIS predictions without wake losses be added to this figure for visual comparison?

   *An additional Subsection 4.1 has been added, comparing FLORIS simulations to LES. This added discussion provides several plots visually comparing differences in accuracy due to wake calculations.*

8. Some of the newly added sentences have typographical errors (e.g. Line 295), I suggest the authors check over them in detail.

   *The newly added sections have been proofread in detail, and typos have been fixed.*

9. Thank you to the authors for including the results of FLORIS without wake losses included. I want to be sure I understand the results you're presenting.

   (a) Comparing Tables 3 and B2:
      i. The wake losses make no difference to turbine specific MAE for velocity of < 5 m/s (expected)
      ii. Including wake losses reduces FLORIS MAE only slightly for 5-11 m/s for the heterogeneous model and has no impact on the homogeneous model (unexpected)
      iii. Including wake losses significantly reduces MAE for >11 m/s where we would not expect significant wake interactions (as the rated wind speed is reached), this is also unexpected. There is no impact for the homogeneous model and more impact for the heterogeneous model.

      *Since the turbines in the observed wind farm are relatively spaced out, it is expected that the wind farm will not show significant wake losses during certain flow conditions. The substantial improvements seen when including wake loss calculations at high wind speeds may be due to the greater influence of turbulence intensity at higher wind speeds, which is factored into the wake model. The LES study presented in subsection 4.1 presents an analysis of a wind farm that has a more densely packed turbine layout and more prominent wake effects for further analysis.*

   (b) Comparing Tables 2 and B1, including wake loss modeling degrades FLORIS's performance in the heterogeneous model.

      *Since the average power predictions of individual turbines within the wind farm do show improvements when including the FLORIS wake*

*calculations, this apparent decrease in total power output accuracy may be a reflection of the self-compensating effect that occurs when taking an overall sum of a wind farm. Some details in wake modeling performance may be misrepresented when the overpredictions and underpredictions between turbines are merged in to one lump sum of error.*

(c) Comparing Tables B1 and B2, why does the heterogeneous model have lower MAE for the farm than homogeneous but higher MAE than homogeneous when MAE is turbine specific?

*These variations in relative model performances may again be due to the effects of turbulence at high wind speeds, which are not calculated when excluding wake losses, and consequently, the turbulence not updating at each turbine based on the wake model as discussed in Annoni et al. (2018); Niayifar and Porté-Agel (2015) A comparison of these two tables shows that the most prominent differences are observed in the higher wind speed ranges, which may be subject to greater uncertainty without wake calculations included.*

(d) Given that including or excluding wake effects in FLORIS seems to have a very small impact on the MAE (there is virtually no impact on homogeneous FLORIS MAE), that seems to indicate to this referee that this is not an ideal test case for a wake model.

*Although the relatively low wake effects in the observed wind farm may be less than optimal, their influence is still prominent enough to be observed in the analysis of wind turbine power predictions, particularly on an individual-turbine basis. Since producing accurate power predictions at individual turbines within a wind farm is so crucial for the development of wind farm controls, wind resource assessment, and many other applications, these results seem to be noteworthy and worth reporting. In further validations of the heterogeneous model, presented in Subsection 4.1, a wind farm with closer turbine interdistances is used to obtain a more thorough indication of the proposed model's capability of modeling wake influence.*

**References**

Annoni, J., Fleming, P., Scholbrock, A., Roadman, J., Dana, S., Adcock, C., Porte-Agel, F., Raach, S., Haizmann, F., and Schlipf, D.: Analysis of control-oriented wake modeling tools using lidar field results, Wind Energy Science, 3, 819–831, 2018.

Brogna, R., Feng, J., Sørensen, J. N., Shen, W. Z., and Porté-Agel, F.: A new wake model and comparison of eight algorithms for layout optimization of wind farms in complex terrain, Applied Energy, 259, 114 189, https://doi.org/10.1016/j.apenergy.2019.114189, 2020.

Fleming, P., King, J., Bay, C. J., Simley, E., Mudafort, R., Hamilton, N., Farrell, A., and Martinez-Tossas, L.: Overview of FLORIS updates, Journal of Physics: Conference Series, 1618, 022 028, https://doi.org/10.1088/1742-6596/1618/2/022028, 2020.

Niayifar, A. and Porté-Agel, F.: A new analytical model for wind farm power prediction, Journal of Physics: Conference Series, 625, 012 039, https://doi.org/10.1088/1742-6596/625/1/012039, 2015.

---

## Author Response (AR3)

**Response to Referee Comment for "Design and analysis of a wake model for spatially heterogeneous flow"**

Corresponding author: Alayna Farrell

March 7, 2021

**Abstract**

The authors would like to thank the referees for their comments through-out the review processes. The authors think that these final revisions have greatly improved the paper and addressed the referees' concerns.

**Referee Report:**

Thanks to the authors for including the FLORIS comparison with LES. These results appear to be more compelling to indicate improvements with the heterogeneous flow wake modeling scheme. I have a few minor comments focusing on improving the clarity of the presented results.

1. There are some typographical errors in the new sentences (e.g. Line 362). I encourage the authors to carefully proofread the paper before finalizing it for publication.

   *The recent additions have been proofread thoroughly.*

2. Eq. 3: The exponent 'p' should be on the cosine, not on the \gamma. $cos^p(\gamma)$.

   *This typo has been fixed.*

3. Line 389, sentence starting with: "The comparison of this metric..."

I assume "this metric" refers to Eq. 4. Figure 14 appears to plot an absolute error for each wind direction which is not the same as MAE defined in Eq. 4, which is an average over all wind directions and would produce only one value, not a plot with wind direction dependency. I believe later, this is defined

as "average mean absolute error." It would improve the clarity of the paper to define the quantities explicitly and to be consistent throughout the results section.

*This paragraph has been revised for clarity, and to more specifically discuss the absolute error depicted in Fig. 14. MAE is meant to only be presented in Fig. 15. The term average MAE refers to the average value of MAE over all of the wind direction cases throughout the study. These distinctions have been further clarified in the paper.*

I would also suggest that Figure 14 does not show that the heterogeneous model provides an improvement in predictions, but Figure 15 clearly does.

*Fig. 14 provides a visual reference for the values reported in Table B1. It is also helpful to compare Fig. 14 to Fig. 15 in order to highlight how the strengths of the heterogeneous model are much more apparent when evaluating power prediction accuracy at the individual turbine level, as opposed to overall farm power output.*

4. Figure 15 is an enlightening result which clearly shows the benefit of turbine specific MAE instead of total wind farm power comparisons and the utility of the proposed model.

   *The authors agree that this figure helps to emphasize the most prominent strengths of the heterogeneous model.*

5. I suggest the authors clarify their quantity of interest notations (e.g. MAE) and descriptions in the results section which appear to be reused and conflicting. In Eq. 4, MAE is defined as the absolute error in wind farm power production averaged over all the wind direction cases. However, in Figure 15, MAE is defined as the absolute error in wind turbine power production averaged over all the turbines in the wind farm.

   *Thank you for pointing out this discrepancy. The descriptions of these metrics have been revised for clarity. It now states that Fig. 14 depicts the absolute error of total farm power output, and Fig. 15 presents the mean absolute error of individual turbines.*

6. Figure 15: Caption states "average absolute error" and y-axis states "mean absolute error."

   *This caption has been changed to match the figure.*

[revised manuscript text omitted]